# ResNet After All?
# Neural ODEs and Their Numerical Solution

**Katharina Ott**
Bosch Center for Artificial Intelligence
Renningen, Germany
University of Tübingen, Germany
`katharina.ott3@de.bosch.com`

**Prateek Katiyar**
Bosch Center for Artificial Intelligence
Renningen, Germany
`prateek.katiyar@de.bosch.com`

**Philipp Hennig**
University of Tübingen
MPI for Intelligent Systems
Tübingen, Germany
`philipp.hennig@uni-tuebingen.de`

**Michael Tiemann**
Bosch Center for Artificial Intelligence
Renningen, Germany
`michael.tiemann@de.bosch.com`

## Abstract

A key appeal of the recently proposed Neural Ordinary Differential Equation (ODE) framework is that it seems to provide a continuous-time extension of discrete residual neural networks. As we show herein, though, trained Neural ODE models actually depend on the specific numerical method used during training. If the trained model is supposed to be a flow generated from an ODE, it should be possible to choose another numerical solver with equal or smaller numerical error without loss of performance. We observe that if training relies on a solver with overly coarse discretization, then testing with another solver of equal or smaller numerical error results in a sharp drop in accuracy. In such cases, the combination of vector field and numerical method cannot be interpreted as a flow generated from an ODE, which arguably poses a fatal breakdown of the Neural ODE concept. We observe, however, that there exists a critical step size beyond which the training yields a valid ODE vector field. We propose a method that monitors the behavior of the ODE solver during training to adapt its step size, aiming to ensure a valid ODE without unnecessarily increasing computational cost. We verify this adaptation algorithm on a common bench mark dataset as well as a synthetic dataset.

## 1 Introduction

The choice of neural network architecture is an important consideration in the deep learning community. Among a plethora of options, Residual Neural Networks (ResNets) (He et al., 2016) have emerged as an important subclass of models, as they mitigate the gradient issues (Balduzzi et al., 2017) arising with training deep neural networks by adding skip connections between the successive layers. Besides the architectural advancements inspired from the original scheme (Zagoruyko & Komodakis, 2016; Xie et al., 2017), recently Neural Ordinary Differential Equation (Neural ODE) models (Chen et al., 2018; E, 2017; Lu et al., 2018; Haber & Ruthotto, 2017) have been proposed as an analog of continuous-depth ResNets. While Neural ODEs do not necessarily improve upon the sheer predictive performance of ResNets, they offer the vast knowledge of ODE theory to be applied to deep learning research. For instance, the authors in Yan et al. (2020) discovered that for specific perturbations, Neural ODEs are more robust than convolutional neural networks. Moreover, inspired by the theoretical properties of the solution curves, they propose a regularizer which improved the robustness of Neural ODE models even further. However, if Neural ODEs are chosen for their theoretical advantages, it is essential that the effective model—the combination of ODE problem and its solution via a particular numerical method—is a close approximation of the true analytical, but practically inaccessible ODE solution.

---

Code: `https://github.com/boschresearch/numerics_independent_neural_odes`

In this work, we study the empirical risk minimization (ERM) problem

$$L_{\mathcal{D}} = \frac{1}{|\mathcal{D}|} \sum_{(x,y) \in \mathcal{D}} l(f(x; w), y) \tag{1}$$

where $\mathcal{D} = \{(x_n, y_n) \mid x_n \in \mathbb{R}^{D_x}, y_n \in \mathbb{R}^{D_y}, n = 1, \ldots, N\}$ is a set of training data, $l :$ $\mathbb{R}^{D_y} \times \mathbb{R}^{D_y} \to \mathbb{R}$ is a (non-negative) loss function and $f$ is a Neural ODE model with weights $w$, i.e.,

$$f = f_d \circ \varphi_T^{f_v} \circ f_u \tag{2}$$

where $f_x, x \in \{d, v, u\}$ are neural networks and $u$ and $d$ denote the upstream and downstream layers respectively. $\varphi$ is defined to be the (analytical) flow of the dynamical system

$$\frac{\mathrm{d}z}{\mathrm{d}t} = f_v(z; w_v), \; z(t) = \varphi_t^{f_v}(z(0)). \tag{3}$$

As the vector field $f_v$ of the dynamical system is itself defined by a neural network, evaluating $\varphi_T^{f_v}$ is intractable and we have to resort to a numerical scheme $\Psi_t$ to compute $\varphi_t$. $\Psi$ belongs either to a class of fixed step methods or is an adaptive step size solver as proposed in Chen et al. (2018). For fixed step solvers with step size $h$ one can directly compute the number of steps taken by the solver $\#steps = Th^{-1}$. We set the final time $T = 1$ for all our experiments. The global numerical error $e_{train}$ of the model is the difference between the true, (unknown), analytical solution of the model and the numerical solution $e_{train} = ||\varphi_T(z(0)) - \Psi_T(z(0))||$ at time $T$. The global numerical error for a given problem can be controlled by adjusting either the step size or the local error tolerance.

Since the numerical solvers play an essential role in the approximation of the solutions of an ODE, it is intuitive to ask: how does the choice of the numerical method affect the training of a Neural ODE model? Specifically, does the discretization of the numerical solver impact the resulting flow of the ODE? To test the effect of the numerical solver on a Neural ODE model, we first train a Neural ODE on a synthetic classification task consisting of three concentric spheres, where the outer and inner sphere correspond to the same class (for more information see Section 2.4). For this problem there are no true underlying dynamics and therefore, the model only has to find some dynamics which solve the problem. We train the Neural ODE model using a fixed step solver with a small step size and a solver with a large step size (see Figure 1 (a) and (b) respectively). If the model is trained with a large step size, then the numerically computed trajectories for the individual Initial Value Problems (IVPs) cross in phase space (see Figure 1 (b)). Specifically, we observe that trajectories of IVPs belonging to different classes cross. This crossing behavior contradicts the expected behavior of autonomous ODE solutions, as according to the Picard-Lindelöf theorem we expect unique solutions to the IVPs. We observe crossing trajectories because the discretization error of the solver is so large that the resulting numerical solutions no longer maintain the properties of ODE solutions.

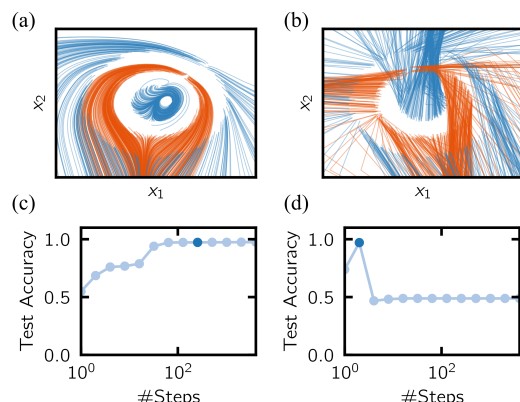

Figure 1: The Neural ODE was trained on a classification task with a small (a) and large (b) step size. (a) and (b) show the trajectories for the two different solvers. The colors of the trajectories indicate the label for each IVP. Panels (c) and (d) show the test accuracy of the Neural ODE solver using different step sizes for testing. (●) indicates the number of steps used for testing are the same as the number of steps used for training. (●) - the number of steps used for testing are different from the number of steps used for training.

We observe that both, the model trained with the small step size and the model trained with the large step size, achieve very high accuracy. This leads us to the conclusion that the step size parameter is not like any other hyperparameter, as its chosen value often does not affect the performance of the model. Instead, the step size affects whether the trained model has a valid ODE interpretation.

Crossing trajectories are not bad per se if the performance is all we are interested. If, however, we are interested in applying algorithms whose success is motivated from ODE theory to, for example, increase model robustness (Yan et al., 2020), then the trajectories must not cross.

We argue that if *any* discretization with similar or lesser discretization error yields the same prediction, the trained model corresponds to an ODE that is qualitatively well approximated by the applied discretization. Therefore, in our experiments we evaluate each Neural ODE model with smaller and larger step sizes than the training step size. We notice that the model trained with the small step size achieves the same level of performance when using a solver with smaller discretization error for testing (Figure 1 (c)). For the model trained with the large step size, we observe a significant drop in performance if the model is evaluated using a solver with a smaller discretization error (see Figure 1 (d)). The reason for the drop in model performance is that the decision boundary of the classifier has adapted to the global numerical error $e_{train}$ in the computed solution. For this specific example, correct classification relies on crossing trajectories as a feature. Therefore, the solutions of solvers with a smaller discretization error are no longer assigned the right class by the classifier and the Neural ODE model is a ResNet model without ODE interpretation.

If we are interested in extending ODE theory to Neural ODE models, we have to ensure that the trained Neural ODE model indeed maintains the properties of ODE solutions. In this work we show that the training process of a Neural ODE yields a discrete ResNet without valid ODE interpretation if the discretization is chosen too coarse. With our rigorous Neural ODE experiments on a synthetic dataset as well as CIFAR10 using both fixed step and adaptive step size methods, we show that if the precision of the solver used for training is high enough, the model does not depend on the solver used for testing as long as the test solver has a small enough discretization error. Therefore, such a model allows for a valid ODE interpretation. Based on this observation we propose an algorithm to find the coarsest discretization for which the model is independent of the solver.

## 2 INTERACTION OF NEURAL ODE AND ODE SOLVER CAN LEAD TO DISCRETE DYNAMICS

We want to study how the Neural ODE is affected by the specific solver configuration used for training. To this end, in our experiments we first train each model with a specific step size $h_{\text{train}}$ (or a specific tolerance $\text{tol}_{\text{train}}$ in the case of adaptive step size methods). For the remainder of this section we will only consider fixed step solvers, but all points made equally hold for adaptive step methods, as shown by our experiments. Post-training, we evaluate the trained models using different step sizes $h_{\text{test}}$ and note how using smaller steps sizes $h_{\text{test}} < h_{\text{train}}$ affects the model performance. We expect that if the model yields good performance in the limiting behavior using smaller and smaller step sizes $h_{\text{test}} \to 0$ for testing, then model should correspond to a valid ODE. For a model trained with a *small* step size, we find that the numerical solutions do not change drastically if the testing step size $h_{\text{test}}$ is decreased (see Figure 1 (c)). But if the step size $h_{\text{train}}$ is beyond a critical value, the model accumulates a large global numerical error $e_{train}$. The decision layer may use these drastically altered solutions as a signal/feature in the downstream computations. In this case, the model is tied to a *specific*, *discrete* flow and the model remains no longer valid in the limit of using smaller and smaller step sizes $h_{\text{test}} \to 0$.

### 2.1 THE TRAJECTORY CROSSING PROBLEM

In this sub-section, we examine the trajectory crossing effect which causes the ODE interpretation to break down. First, we look at the numerically computed trajectories in phase space of a Neural ODE model trained with a very large step size of $h_{\text{train}} = 1/2$ (see Figure 1 (b)). A key observation is that the trajectories cross in phase space. This crossing happens because the step size $h_{\text{train}}$ is much bigger than the length scale at which the vector field changes, thus missing "the curvature" of the true solution. Specifically, we observe that the model exploits this trajectory crossing behavior as a feature to separate observations from different classes. This is a clear indication that these trajectories do not approximate the true analytical solution of an ODE, as according to the Picard-Lindelöf theorem (Hairer et al., 1993, § 1.8), solutions of first order autonomous ODEs do not cross in phase space. Since the numerical solutions using smaller step sizes $h_{\text{test}} < h_{\text{train}}$ no longer maintain the crossing trajectory feature, the classifier cannot separate the data with the learned vector field (see Figure 2).

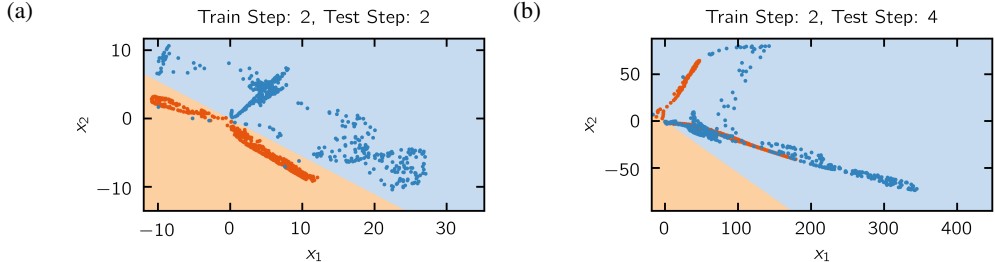

Figure 2: Output of the Neural ODE block for different step sizes The mode is the same as in Figure 1, trained with a step size of $1/2$. The points indicate the output of the Neural ODE block, the color of the points shows their true label, and the light color in the background indicates the label assigned by the classifier to this region. The model was tested with a step size of $1/2$ in (a) and $1/4$ in (b). Axis were scaled such that the final locations of all test data points after passing through the ODE block are shown.

## 2.2 LADY WINDERMERE'S FAN

In cases where trajectory crossings do *not* occur, other, more subtle effects can also lead to a drop in performance in the limit of using smaller and smaller test step sizes $h_{\text{test}} \to 0$. The compound effect of local numerical error leads to a biased global error which is sometimes exploited as a feature in downstream blocks. This effect how the local error gets accumulated into global error was coined as Lady Windermere's Fan in Hairer et al. (1993, § 1.7).

To understand these effects we introduce an example based on the XOR problem $D = \{((0,0) \mapsto 0), ((1,1) \mapsto 0), ((0,1) \mapsto 1), ((1,0) \mapsto 1)\}$. This dataset cannot be classified correctly in 2D with a linear decision boundary (Goodfellow et al., 2016, § 1.2). Therefore, we consider the ODE

$$z'(t) = \begin{pmatrix} \alpha & 1 \\ -\gamma ||z||^\delta & \beta \end{pmatrix} z. \tag{4}$$

The qualitative behavior of the analytical flow are increasing ellipsoids with ever increasing rotational speed. We chose this problem as an example based on the knowledge that the precision of a solver influences how the rotational speed of the ellipsoids is resolved. Therefore, this problem is useful in illustrating how the numerical accuracy of the solver can affect the final solution.

Fig. 3 depicts the numerical solution of this flow with one set of fixed parameters and different step sizes $h = 10^{-2.5}, 10^{-3.5}$. For both step sizes we do not observe crossing trajectories, but the final solutions differ greatly. For $h = 10^{-2.5}$ the numerical flow produces a transformation in which the data points can be separated linearly. But for the smaller step size of $h = 10^{-3.5}$, the numerical solution is no longer linearly separable. The problem here is that the numerical solution using the larger step size is not accurate enough to resolve the rotational velocity. For each step the local error gets accumulated into the global error. In Figure 3 (a), the accumulation of error in the numerical solution results in a valid feature for a linear decision (classification) layer. The reason for this is that the global numerical errors $e_{train}$ are biased. We define as the *fingerprint* of the method the structure in the global numerical error. The decision layer then adapts to this method specific fingerprint. How this fingerprint affects the performance of the model when using smaller step size $h_{\text{test}}$ is dependent on two aspects. First, does the data remain separable when using smaller step sizes $h_{\text{test}} < h_{\text{train}}$? If not, we will observe a significant drop in performance. Second, how sensitive is the decision layer to changes in the solutions and how much do the numerical solutions change when decreasing the test step size $h_{\text{test}} \to 0$? Essentially, the input sensitivity of the downstream layer should be less than the output sensitivity of $h_{\text{test}} < h_{\text{train}}$. For the decision layer, there should exist a sensitivity threshold $d$ such that $f_d(z(T) + \delta) = f_d(z(T)) \; \forall ||\delta|| < d$. Thus, if two solvers compute the same solution up to $\delta$, the classifier identifies these solutions as the same class and the result of the model is not affected by the interchanging these solvers.

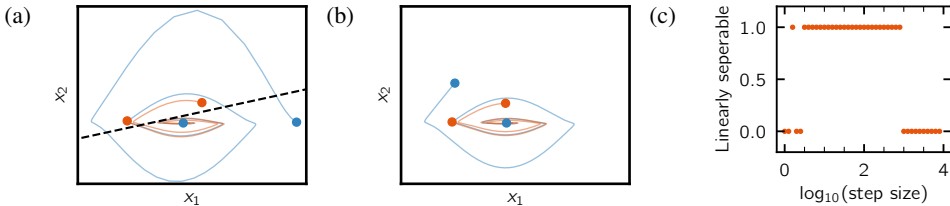

Figure 3: Solutions to Eq. 4 using Euler's method with step size $h = 10^{-2.5}$ (a) and $h = 10^{-3.5}$ (b). The points indicate where each IVP ends up in phase space and the color indicates the class the solution belongs to. The trajectories taken by the numerical solver are shown in the color indicating the respective class. In (a) the black dotted line indicates one possibility how to separate the data linearly. (c) shows for which step sizes the solution is linearly separable (1) or not (0).

## 2.3 DISCUSSION

We have just described two effects, trajectory crossing and Lady Windermere's fan, which can lead to a drop in performance when the model is tested with a different solver. The trajectory crossing effect is a clear indication that the model violates ODE-semantics and it is not valid to apply ODE-theory to this model. But even if we do not observe trajectory crossing for some step size $\tilde{h}$ we are not guaranteed to not observe trajectory crossings for all $h < \tilde{h}$ (see Supplementary Material Section A for an example). On the other other hand, note that the occurrence of Lady Windermere's fan as a downstream feature *does not* violate an ODE interpretation of the numerical flow. However, we argue that a pronounced Lady Windermere's fan should still be avoided as a downstream feature because the feature vanishes in the continuous-depth limit leading the Neural ODE model ad absurdum. Additionally, distinguishing Lady Windermere's fan from the trajectory crossing problem is hard without visualization as both effects lead to a drop in performance if a solver with higher numerical accuracy is used for testing. However, there exists a critical step size/tolerance $\theta_{crit}$ where convergence is close to floating point accuracy and both effects vanish in the limit. As a first step we propose to check how robust the model is with respect to the step size/tolerance to ensure that the resulting model is in a regime where ODE-ness is guaranteed and therefore one can apply reasoning from ODE theory to the model.

The current implementation of Neural ODEs does not ensure that the model is driven towards continuous semantics as there are no checks in the gradient update ensuring that the model remains a valid ODE nor are there penalties in the loss function if the Neural ODE model becomes tied to a specific numerical configuration. An interesting direction would also be to regularize the Neural ODE block towards continuous semantics. One idea is to restrict the Lipschitz constant to below 1, as proposed by Behrmann et al. (2018) for ResNets which avoids crossing trajectories.

## 2.4 EXPERIMENTS

For our experiments, we introduce a classification task based on the concentric sphere dataset proposed by Dupont et al. (2019) (see Supplementary Material Figure 8). Whether this dataset can be fully described by an autonomous ODE, is dependent on the degrees of freedom introduced by combining the Neural ODE with additional downstream (and upstream) layers.

In this subsection, we present results from the experiments performed on Sphere2 and CIFAR10 datasets using fixed step and adaptive step solvers. For additional results on MNIST we refer to the Supplementary Material Section B. The aim of these experiments is to analyze the dynamics of Neural ODEs and show its dependence on the specific solver used during training by testing the model with a different solver configuration. In the main experiments presented in the paper, we choose to back-propagate through the numerical solver. The results pertaining to the adjoint method (Chen et al., 2018) are provided in the Supplementary Material Section B. For all our experiments, we do not use an upstream block $f_u$ similar to the architectures proposed in Dupont et al. (2019). Additionally, we decided to only use a *single* ODE block and a simple architecture for the classifier. We chose such an architectural scheme to maximize the modeling contributions of the ODE block. We do not expect to achieve state-of-the-art results with this simple architecture but we expect our results to remain valid for more complicated architectures.

For training the Neural ODE with fixed step solvers, Euler's method and a 4th order Runge–Kutta (rk4) method were used (descriptions of these methods can be found in Hairer et al. (1993)). The trained Neural ODE was then tested with different step sizes and solvers. For a Neural ODE trained with Euler's method, the model was tested with Euler's method, the midpoint method and the rk4 method. The testing step size was chosen as a factor of 0.5, 0.75, 1, 1.5, and 2 of the original step size used for training. For rk4, we only tested using the rk4 method with different step sizes. Likewise, the adaptive step solver experiments were performed using Fehlberg21 and Dopri54. The models were trained and tested using different tolerances and solvers. The models trained using Fehlberg21 were tested using Fehlberg21 and Dopri54, whereas the models trained using Dopri54 were only tested using Dopri54. The testing tolerance was chosen as a factor of 0.1, 1, and 10 of the original tolerance used for training. Here we do not show the results for all training step sizes and tolerances, but reduced the data to maintain readability of the plots (for plots showing all the data see the Supplementary Material Section B). We report an average over five runs, where we used an aggregation of seeds for which the Neural ODE model trained successfully (the results for all seeds are disclosed in the Supplementary Material). We did not tune all hyper-parameters to reach best performance for each solver configuration. Rather, we focused on hyper-parameters that worked well across the entire range of step sizes and tolerances used for training (see Supplementary Material for the choice of hyper parameters and the architecture of the Neural ODE).

As shown in Figure 4, when training and testing the model with the same step size (a), (b) or the same tolerance (c), (d), the test accuracy does not show any clear dependence on the step size or tolerance respectively. Since we did not tune the learning rate for each step size/tolerance, any visible trends could be due to this choice. Indeed, many different solver configurations work well in practice, but only for small enough step sizes/tolerances the model represents a valid ODE. On both datasets, we observe similar behavior for dependence of the test accuracy on the test solver: when using large step sizes/tolerances for training, the Neural ODE shows dependence on the solver used for testing. But there exists some critical step size/tolerance below which the model shows no clear dependence on the test solver as long as this test solver has equal or smaller numerical error than the solver used for training (as seen in Figure 4). For additional experimental results we refer the reader to the Supplementary Material Section B.1.

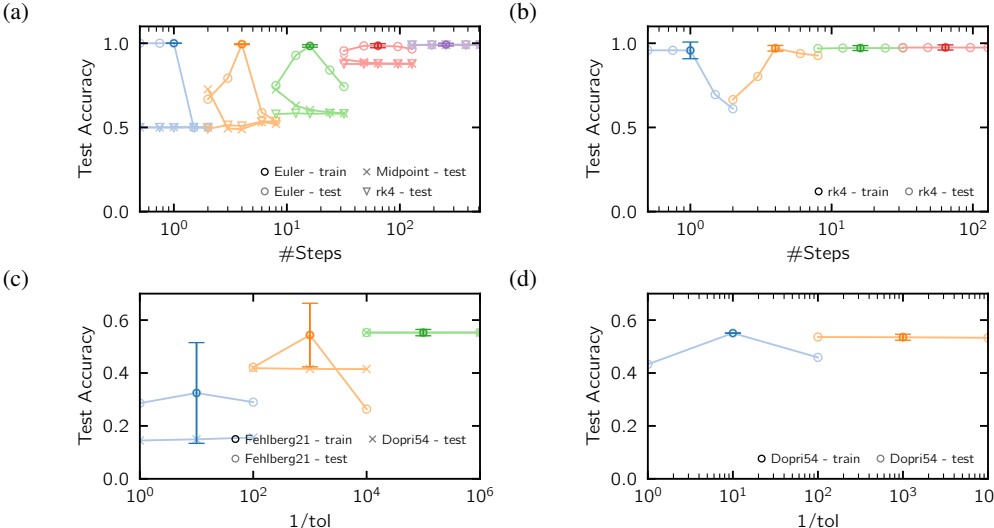

Figure 4: A Neural ODE was trained with different step sizes on Sphere2 (a), (b) and different tolerances on CIFAR10 (c), (d). The model was tested with different solvers and different step sizes/tolerances. In (a) the model was trained using Euler's method. Dark circles indicate that the same solver is used for training and testing. Light data indicates different step sizes used for testing. Circles correspond to Euler's method, cross to the midpoint method and triangles to rk4. In (b) a 4th order Runge Kutta methods was used for training and testing. In (c) the model was trained using the Fehlberg21 method. Circles correspond to Fehlberg21 method, cross to the Dopri54 method. In (d) Dopri54 was used for training (dark circles) and testing (light circles).

The aforementioned dynamics of Neural ODE were also verifiable in the adaptive step solver experiments (see Figure 4, (c) and (d)). In this case, the trained model's test accuracy was dependent on the configuration of the test solver below a critical tolerance value. For additional results on the Sphere2 dataset we refer the reader to the Supplementary Material Section B.2.

## 3 ALGORITHM FOR STEP SIZE ADAPTATION

Although Neural ODEs achieve good accuracy for a large variety of solver configurations, if theoretic results of ODEs are to be applicable to Neural ODEs, it is paramount to find a solution corresponding to an ODE flow. To ensure this, we propose an algorithm that checks whether the Neural ODE remains independent of the specific train solver configuration and adapts the step size for fixed step solver and the tolerance for adaptive solvers if necessary. The proposed algorithm tries to find solver settings throughout training which keep the number of function evaluations small, while maintaining continuous semantics.

---

**Algorithm 1:** Step and tolerance adaptation algorithm

---

**Inputs** $\epsilon$, train_solver, test_solver, model;
**while** *Training* **do**
    batch = draw_batch(data);
    logits = model.forward_pass(batch, train_solver($\epsilon$));
    loss = model.calculate_loss(logits);
    train_solver_acc = model.calculate_acc(logits);
    **if** *Iteration % 50 == 0* **then**
        logits = model.do_forward_pass(batch,
         test_solver($\epsilon$));
        test_solver_acc = model.calculate_acc(logits);
        **if** $|train\_solver\_acc\text{-}test\_solver\_acc| > 0.1$ **then**
            $\epsilon = 0.5\ \epsilon$;
        **else**
            $\epsilon = 1.1\ \epsilon$;
        **end**
    **end**
    model.update(loss);
**end**

---

It is important to note that adaptive step size methods with one fixed tolerance parameter do not solve this issue, as embedded methods can severely underestimate the local numerical error if the vector field is not sufficiently smooth (Hairer et al., 1993). In contrast to the common application of such methods, in the case of Neural ODEs we cannot choose the appropriate solver and tolerance for a given problem as the vector field of the Neural ODE block is changing throughout training. While there always exists a low enough tolerance such that the adaptation issue does not occur, this low enough tolerance may be prohibitively small in practice and is certainly leaving runtime efficiency on the table.

So far, there does not exist any other algorithm that we are aware of which solves the issue. The aim of the proposed algorithms is not to achieve state of the art results, but rather be a first step towards ensuring that trained Neural ODE models can be viewed independently of the solver configuration used for training. Here we will describe the algorithm for the fixed step solvers, which shows promising results for an equivalend algorithm fo datptive methods see in the Supplementary Material Section C). Pseudo-code for both settings is presented in Alg. 1.

First the algorithm has to initialize the accuracy parameter $\epsilon$, which corresponds to the step size $h$ for fixed step solvers and the tolerance for adaptive step size solvers. The initial step size is chosen according to an algorithm described by Hairer et al. (1993)[p. 169] which ensures that the Neural ODE chooses an appropriate step size for all neural networks and initializations. In our experiments we found that the initial step size suggested by the algorithm is not too small, which makes the algorithm useful in practice. The Neural ODE starts training with the proposed accuracy parameter $\epsilon$. After a predefined number of iterations (we chose $k = 50$), the algorithm checks whether the model can still be interpreted as a valid ODE: the accuracy is calculated over one batch with the train solver and with a test solver, where the test solver has a smaller discretization error than the train solver. To ensure that the test solver has a smaller discretization error than the train solver, the accuracy parameter $\epsilon$ is adjusted for testing if necessary (see Supplementary Material Section C for details). If test and train solver show a significant difference in performance, we decrease the accuracy parameter and let the model train with this accuracy parameter to regain valid ODE dynamics. If performance of test solver and train solver agree up to a threshold, we cautiously increase the accuracy parameter.

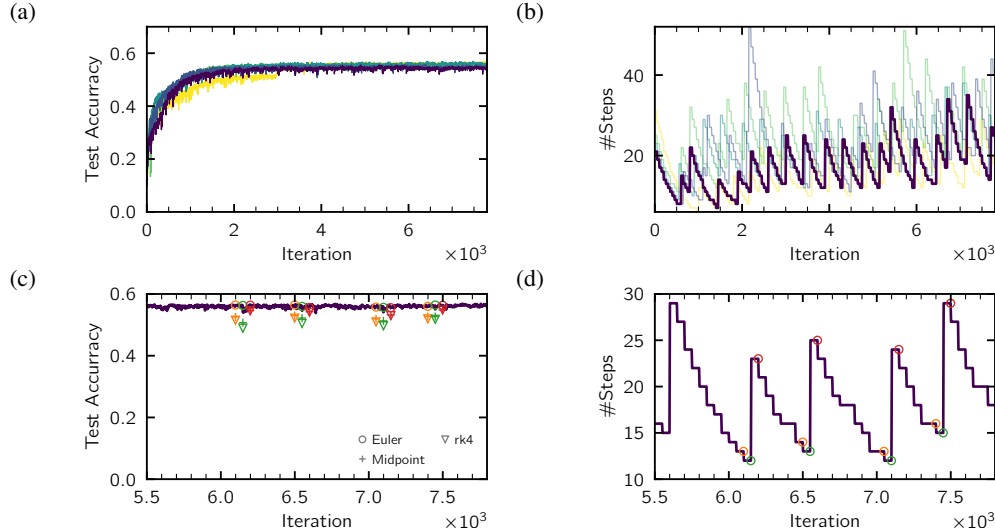

Figure 5: Using the step adaptation algorithm for training on CIFAR10 (a), (b). (a) shows the test accuracy over the course of training for five different seeds. (b) shows the number of steps chosen by the algorithm over the course of training. (c) shows the test accuracy. At certain points in time (also marked in (d)), the model is evaluated with solvers of smaller discretization error (orange green and red data points). Triangles correspond to a 4th order Runge Kutta method, crosses to the midpoint method. (d) shows the number of steps chosen by the algorithm.

Unlike in ODE solvers, the difference between train and test accuracy does not tell by how much the step size needs to be adapted, so we choose some constant multiplicative factor that works well in practice (see Algorithm 1 for a simplified version and the Supplementary Material for details). The algorithm was robust against small changes to the constants in the algorithm.

## 3.1 EXPERIMENTS

We test the step adaptation algorithm on two different datasets: the synthetic dataset and on CIFAR10 (see Supplementary Material for addational results). We use Euler's method as the train solver and the midpoint method as the test solver (addtional configurations are found in the Supplementary material). On all datasets, we observe that the number of steps taken by the solver fluctuate over the course of training (see Figure 5). The reason for such a behavior is that the algorithm increases the step size until the step size is too large and training with this step size leads to an adaptation of the vector field to this particular step size. Upon continuing training with a smaller step size, this behavior is corrected (see Figure 5 (c) and (d)) and the algorithm starts increasing the step size again. To compare the results of the step adaptation algorithm to the results of the grid search, we detail accuracy as well as number of average function evaluations (NFE) per iteration. For the grid search, we determine the critical number of steps using the same method as in the step adaptation algorithm. We report the two step sizes above and below the critical step size which were part of the grid search. For the step adaptation algorithm we calculate the NFE per iteration by including all function evaluations over course of training (see Table 1). The achieved accuracy and step size found by our algorithm is on par with the smallest step size above the critical threshold thereby eliminating the need for a grid search.

## 4 RELATED WORK

The connections between ResNets and ODEs have been discussed in E (2017); Lu et al. (2018); Haber & Ruthotto (2017); Sonoda & Murata (2019). The authors in Behrmann et al. (2018) use similar ideas to build an invertible ResNet. Likewise, additional knowledge about the ODE solvers can be exploited to create more stable and robust architectures with a ResNet backend (Haber & Ruthotto, 2017; Haber et al., 2019; Chang et al., 2018; Ruthotto & Haber, 2019; Ciccone et al., 2018; Cranmer et al., 2020; Benning et al., 2019).

| Data set | Grid search | | Step adaptation algorithm | |
|---|---|---|---|---|
| | NFE | Accuracy | NFE | Accuracy |
| Concentric spheres 2d | 65-129 | $98.7 \pm 1.0\%$ | 100.5 | $98.9 \pm 0.6\%$ |
| Cifar10 | 17-33 | $54.7 \pm 0.3\%$ | 21.9 | $55.0 \pm 0.8\%$ |

Table 1: Results for the accuracy and the number of function evaluations to achieve time continuous dynamics using a grid search and the proposed step adaptation algorithm. For the grid search, we report the accuracy of the run with the smallest step size above the critical threshold.

Continuous-depth deep learning was first proposed in Chen et al. (2018); E (2017). Although ResNets are universal function approximators (Lin & Jegelka, 2018), Neural ODEs require specific architectural choices to be as expressive as their discrete counterparts (Dupont et al., 2019; Zhang et al., 2019a; Li et al., 2019). In this direction, one common approach is to introduce a time-dependence for the weights of the neural network (Zhang et al., 2019c; Avelin & Nyström, 2020; Choromanski et al., 2020; Queiruga et al., 2020). Other solutions include, novel Neural ODE models (Lu et al., 2020; Massaroli et al., 2020) with improved training behavior, and variants based on kernels (Owhadi & Yoo, 2019) and Gaussian processes (Hegde et al., 2019). Adaptive ResNet architectures have been proposed in Veit & Belongie (2018); Chang et al. (2017). The dynamical systems view of ResNets has lead to the development of methods using time step control as a part of the ResNet architecture (Yang et al., 2020; Zhang et al., 2019b). Thorpe & van Gennip (2018) show that in the deep limit the Neural ODE block and its weights converge. This supports our argument for the existence of a critical step size. Weinan et al. (2019) and Bo et al. (2019) show the theoretical implications and advantages a continuous formulation ResNet models has.

Gusak et al. (2020) and Zhuang et al. (2020) observe a drop in performance when changing to numerically more precise solvers. In a similar vein as our work, Queiruga et al. (2020) study how the solver influences the Neural ODE model, showing that a model trained with Euler's method can have significantly lower performance when tested with a higher order solver. To avoid this issue, they propose to use higher order solvers for training Neural ODEs.

## 5 CONCLUSION

We have shown that the step size of fixed step solvers and the tolerance for adaptive methods used for training Neural ODEs impacts whether the resulting model maintains properties of ODE solutions. As a simple test that works well in practice, we conclude that the model only corresponds to a continuous ODE flow, if the performance does not depend on the exact solver configuration. We illustrated that the reasons for the model to become dependent on a specific train solver configuration are the use of the bias in the numerical global errors as a feature by the classifier, and the sensitivity of the classifier to changes in the numerical solution. We have verified this behavior on CIFAR10 as well as a synthetic dataset using fixed step and adaptive methods. Based on these observations, we developed step size and tolerance adaptation algorithms, which maintain a continuous ODE interpretation throughout training. For minimal loss in accuracy and computational efficiency, our step adaptation algorithm eliminates a massive grid search. In future work, we plan to eliminate the oscillatory behavior of the adaptation algorithm and improve the tolerance adaptation algorithm to guarantee robust training on many datasets.

ACKNOWLEDGMENTS

The authors thank Andreas Look, Nicholas Krämer, Kenichi Nakazato and Sho Sonoda for helpful discussions. PH gratefully acknowledges financial support by the German Federal Ministry of Education and Research (BMBF) through Project ADIMEM (FKZ 01IS18052B); the European Research Council through ERC StG Action 757275 / PANAMA; the DFG Cluster of Excellence Machine Learning - New Perspectives for Science, EXC 2064/1, project number 390727645; the German Federal Ministry of Education and Research (BMBF) through the Tübingen AI Center (FKZ: 01IS18039A); and funds from the Ministry of Science, Research and Arts of the State of Baden-Württemberg.

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

## A    CROSSING TRAJECTORIES

In the main text we describe how trajectory crossings are an indication that the vector field has adapted to a specific solver and such a vector field then has no meaningful interpretation in the continuous setting. In this section we want to answer whether the following statement is true: If for a step size $\tilde{h}$ we do not observe crossing trajectories then for all $h < \tilde{h}$ we will not observe crossing trajectories. To study this problem in more detail we introduce a vector field $f : [0.0, \infty) \times [0.1, \infty) \to [0, \infty) \times [0, \infty)$

$$\frac{\mathrm{d}}{\mathrm{d}t} \begin{bmatrix} x & y \end{bmatrix} = f(x, y), \tag{5}$$

$$f(x, y) = \begin{cases} \begin{bmatrix} 1 - 2(x - 1) & \frac{2(x-1)}{y} \end{bmatrix}, & \text{if } x \in (1, 1.25] \\ \begin{bmatrix} 0.5 & \frac{0.5}{y} \end{bmatrix}, & \text{if } x \in (1.25, 1.75] \\ \begin{bmatrix} 0.5 + 2(x - 1.75) & \frac{0.5 - 2(x-1.75)}{y} \end{bmatrix}, & \text{if } x \in (1.75, 2] \\ \begin{bmatrix} 1 & 0 \end{bmatrix}, & \text{else.} \end{cases} \tag{6}$$

We chose this vector field because only for the region $x \in (1, 2]$ the curvature of the vector field changes (see Figure 6 for a visualization of the vector field). The described vector field is Lipschitz continuous on $[0.0, \infty) \times [0.1, \infty)$ and autonomous, therefore the Picard-Lindelöf theorem applies and the true solutions to the ODE do not cross in phase space.

We use Euler's method with different step sizes to solve the ODE. First we only look at the IVPs where $x(t = 0) = 0$ (shown in blue in Figure 7). Each IVP can be thought of as a data point in a dataset. For the large step size $h = 1$ we do not observe crossing trajectories (see Figure 7 (a)). The numerical solver fails to resolve the change in curvature in the vector field. If we decrease the step size to $h = 1/2$ we observe crossing trajectories (see Figure 7 (b)). This clearly shows that not observing crossing trajectories for some $\tilde{h}$ does not give us any information about what will happen for $h < \tilde{h}$. But since we know that the numerical solvers converge to the true solution, we can make the following statement: For a given set of IVPs there exists a step size $h^*$ such that for all $h < h^*$ we do not observe crossing trajectories. And indeed if we choose an even smaller step size we no longer observe crossing trajectories (see Figure 7 (c)).

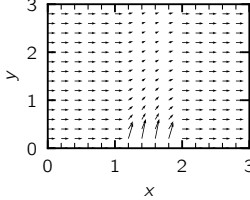

Figure 6: Visualization of the vector field defined in Eq. (6)

We also want to emphasize that observed behavior for the different step sizes is dependent on the set of IVPs. To illustrate this point we add additional IVPs to our original dataset, where for the new IVP $x(t = 0) = 1.4$ (the additional IVPs are shown in orange in Figure 7). Now we also observe crossing trajectories for the large step size $h = 1$. This shows that whether we observe crossing trajectories or not is not only dependent on the step size but also on the set of IVPs we choose.

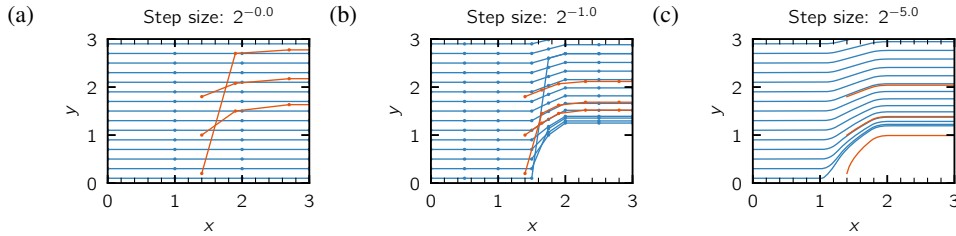

Figure 7: Numerical solutions to Eq.6 using Euler's method. In blue is the set of IVPs for which $x(0) = 0$ and in orange is an additional set of IVPs for which $x(t = 0) = 1.5$. The ODE is solved using different step size, $h = 1$ in (a), $h = 2^{-1}$ in (b), and $h = 2^{-5}$ in (c).

# B EXPERIMENTAL RESULTS

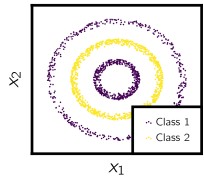

Figure 8: The concentric sphere dataset (Sphere2).

In the main text we do not plot the results for all the training step sizes/ tolerances but only for every second training step size/tolerance to improve the clarity of the plots. Here we now include the plots showing all training runs and we include additional results for all datasets.

## B.1 RESULTS FOR FIXED STEP SOLVERS

In addition to the concentric sphere dataset in 2 dimensions (see Figure 8), we introduce higher dimensional versions of the concentric sphere dataset, namely 3, 10, and 900 dimensional versions.

In this section we present the results for fixed step solvers. The model is trained with Euler's method or a 4th order Runge Kutta method (rk4) with different step sizes. If Euler's method is used for training then the model is tested with Euler's method, the Midpoint method and rk4. If rk4 is used for training then the model is only tested with rk4.

We train Neural ODE models on CIFAR10 (see Figure 9), on MNIST (see Figure 10), on the 2 dimensional concentric sphere dataset (see Figure 11), on the 3 dimensional concentric sphere dataset (see Figure 12), on the 10 dimensional concentric sphere dataset (see Figure 13), and on the 900 dimensional concentric sphere dataset (see Figure 14).

For all datasets we observe that if the model is trained with a large step size then there is a drop in performance if a solver with smaller numerical error is used for testing. But there exists a training step size above which using a solver with a higher numerical accuracy does not lead to a drop in performance. The observations support our claims made in the main text.

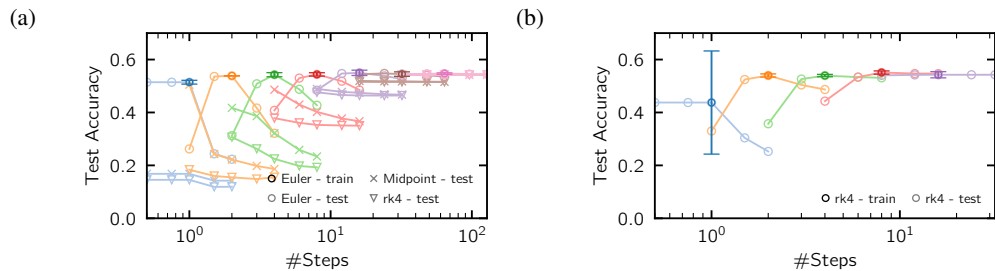

Figure 9: A Neural ODE was trained with different step sizes (plotted in different colors) on CIFAR10 (a), (b) . The model was tested with different solvers and different step sizes. In (a) the model was trained using Euler's method. Results obtained by using the same solver for training and testing are marked by dark circles. Light data indicated different step sizes used for testing. Circles correspond to Euler's method, cross to the midpoint method and triangles to a 4th order Rung Kutta method. In (b) a 4th order Runge Kutta methods was used for training (dark circles) and testing (light circles).

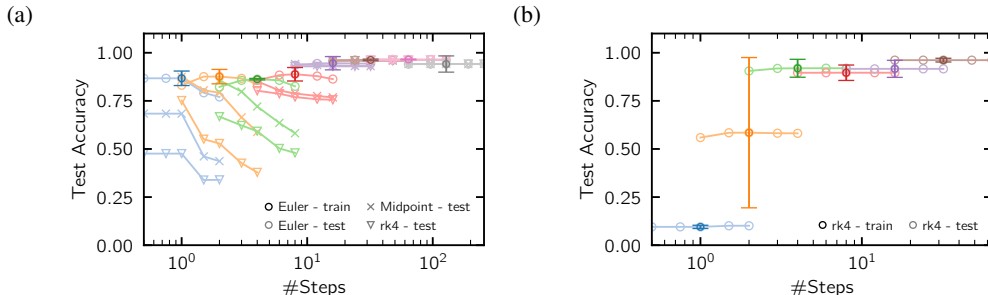

Figure 10: A Neural ODE was trained with different step sizes (plotted in different colors) on MNIST (a), (b) . The model was tested with different solvers and different step sizes. In (a) the model was trained using Euler's method. Results obtained by using the same solver for training and testing are marked by dark circles. Light data indicated different step sizes used for testing. Circles correspond to Euler's method, cross to the midpoint method and triangles to a 4th order Rung Kutta method. In (b) a 4th order Runge Kutta methods was used for training (dark circles) and testing (light circles).

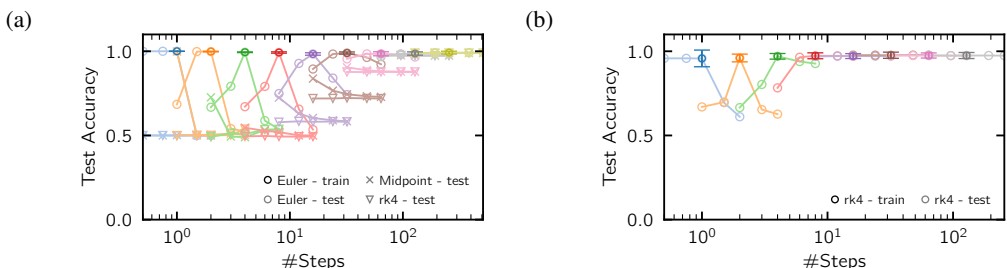

Figure 11: A Neural ODE was trained with different step sizes (plotted in different colors) on the 3 dimensional concentric sphere dataset (a), (b). The model was tested with different solvers and different step sizes. In (a) the model was trained using Euler's method. Results obtained by using the same solver for training and testing are marked by dark circles. Light data indicated different step sizes used for testing. Circles correspond to Euler's method, cross to the midpoint method and triangles to a 4th order Rung Kutta method. In (b) a 4th order Runge Kutta methods was used for training (dark circles) and testing (light circles).

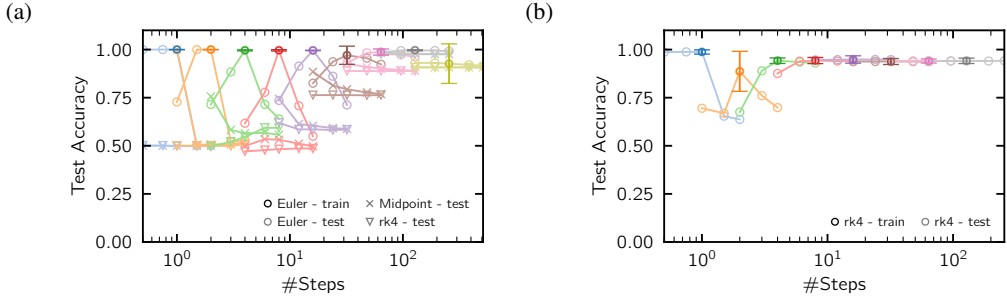

Figure 12: A Neural ODE was trained with different step sizes (plotted in different colors) on the 3 dimensional concentric sphere dataset (a), (b). The model was tested with different solvers and different step sizes. In (a) the model was trained using Euler's method. Results obtained by using the same solver for training and testing are marked by dark circles. Light data indicated different step sizes used for testing. Circles correspond to Euler's method, cross to the midpoint method and triangles to a 4th order Rung Kutta method. In (b) a 4th order Runge Kutta methods was used for training (dark circles) and testing (light circles).

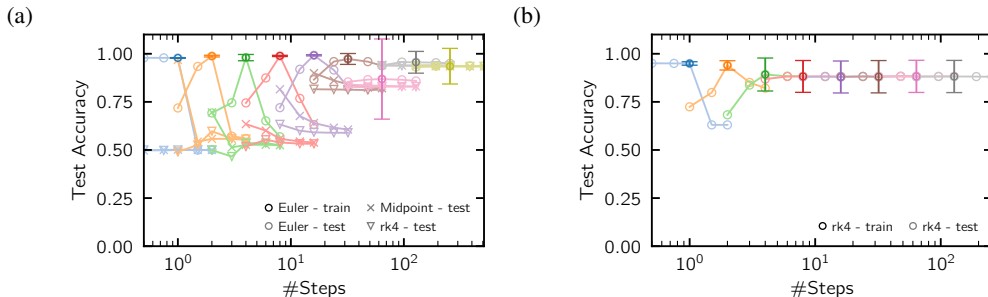

Figure 13: A Neural ODE was trained with different step sizes (plotted in different colors) on the 10 dimensional concentric sphere dataset (a), (b). The model was tested with different solvers and different step sizes. In (a) the model was trained using Euler's method. Results obtained by using the same solver for training and testing are marked by dark circles. Light data indicated different step sizes used for testing. Circles correspond to Euler's method, cross to the midpoint method and triangles to a 4th order Rung Kutta method. In (b) a 4th order Runge Kutta methods was used for training (dark circles) and testing (light circles).

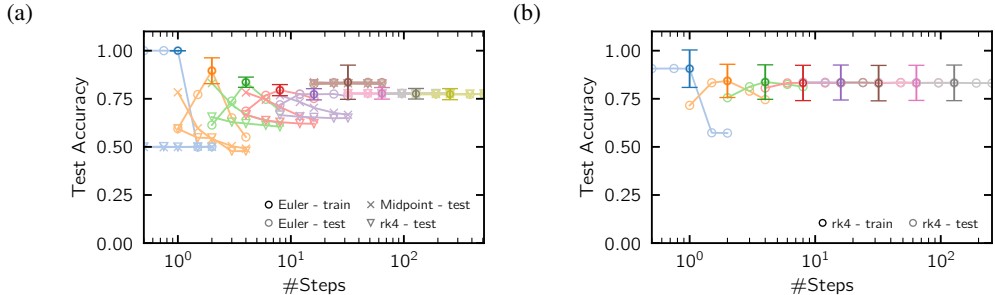

Figure 14: A Neural ODE was trained with different step sizes (plotted in different colors) on the 900 dimensional concentric sphere dataset (a), (b). The model was tested with different solvers and different step sizes. In (a) the model was trained using Euler's method. Results obtained by using the same solver for training and testing are marked by dark circles. Light data indicated different step sizes used for testing. Circles correspond to Euler's method, cross to the midpoint method and triangles to a 4th order Rung Kutta method. In (b) a 4th order Runge Kutta methods was used for training (dark circles) and testing (light circles).

## B.2 RESULTS FOR ADAPTIVE SOLVERS

In this section we present the results for adaptive step size solvers. The model is trained with Fehlberg21 or Dopri54 with different tolerances. If Fehlberg21 is used for training then the model is tested with Fehlberg21 and Dopri54. If Dopri54 is used for training then the model is only tested with Dopri54.

We train the Neural ODE models on CIFAR10 (see Figure 15) on MNIST (see Figure 16) and on the Sphere2 dataset (see Figure 17). For CIFAR10 we use backpropagation through the numerical solver to calculate the gradients. For Sphere2 we use backpropagation through the numerical solver as well as the adjoint method described in Chen et al. (2018) to calculate the gradients.

We observe that the models trained with a relatively large tolerance show a drop in performance if the models are tested with a solver with lower numerical error. We oberserve this behavior for all data sets and also for the adjoint method (see Figure 17 (c) and (d)), the only exception is MNIST trained with Dopri54 which we attribute to the high order of the solver and the low critical step size on MNIST. If the model is trained with a small tolerance then there is no drop in performance if the model is tested with a solver with lower numerical error. We note that the Sphere2 dataset trained with Dopri54 (Figure 17 (b)) shows decreasing performance for smaller tolerances. This might be due to the model taking a lot of steps and therefore, the gradient provided by backpropagation might not be accurate enough. Additionally, tuning the hyperparameters for specific tolerances might improve the performance. For the model trained on Sphere2 using the adjoint method we observe that for large tolerances the model achieves relatively low test accuracy. At large tolerances the model takes relatively large and inaccurate steps. Therefore, the backward solutions of the ODE differs from the forward solve and no valid gradient information is provided to the optimizer. Overall the performance on MNIST is lower than with fixed step solvers, even though we specifically tuned the learning rate and optimizer.

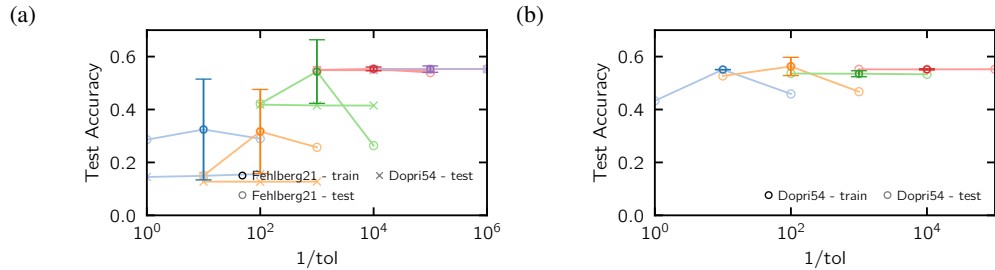

Figure 15: A Neural ODE was trained with different tolerances (plotted in different colors)on CIFAR10 (a), (b). The model was tested with different solvers and different tolerances. In (a) the model was trained using the Fehlberg21 method. Results obtained by using the same solver for training and testing are marked by dark circles. Light data indicated different step sizes used for testing. Circles correspond to Fehlberg21 method, cross to the Dopri54 method. In (b) Dopri54 was used for training (dark circles) and testing (light circles).

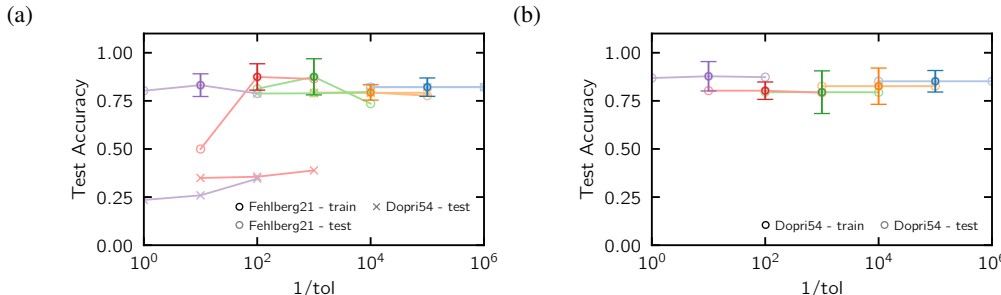

Figure 16: A Neural ODE was trained with different tolerances (plotted in different colors)on MNIST (a), (b). The model was tested with different solvers and different tolerances. In (a) the model was trained using the Fehlberg21 method. Results obtained by using the same solver for training and testing are marked by dark circles. Light data indicated different step sizes used for testing. Circles correspond to Fehlberg21 method, cross to the Dopri54 method. In (b) Dopri54 was used for training (dark circles) and testing (light circles).

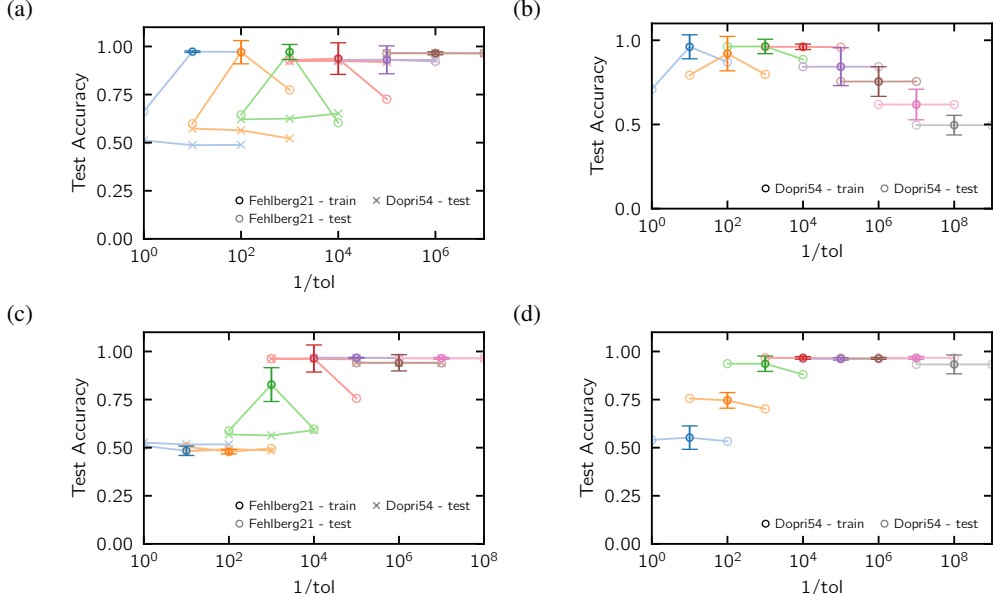

Figure 17: A Neural ODE was trained with different tolerances (plotted in different colors)on Sphere2 (a), (b) and on Sphere2 using the adjoint method (c), (d). The model was tested with different solvers and different tolerances. In (a), (c) the model was trained using the Fehlberg21 method. Results obtained by using the same solver for training and testing are marked by dark circles. Light data indicated different step sizes used for testing. Circles correspond to Fehlberg21 method, cross to the Dopri54 method. In (b), (d) Dopri54 was used for training (dark circles) and testing (light circles).

## C STEP SIZE AND TOLERANCE ADAPTATION ALGORITHM

### C.1 STEP ADAPTATION ALGORITHM

Here we describe the full step adaptation algorithm with additional details omitted in the main text for clarity. The algorithm queries whether the suggested step size has surpassed the length of the time interval over which the ODE is integrated. If this is the case, the step size is reduced to the size of the time interval. This check is necessary to avoid infinitely increasing the step size. The algorithm does not increase the step size if accuracy of the model tested with the train solver and the proposed step size is past the threshold. We do not want the model to continue training with a too large step size leading to discrete dynamics.

Additionally, it is important to ensure that the test solver is more accurate than the train solver. For our experiments we use as a combination of train and test solver the following pairs (Euler, Midpoint), (Euler, rk4), (Midpoint, rk4). To ensure that the test solver achieves a smaller numerical error, the test solver reduces the step size for testing if the training step size is too large. In our algorithm we require that the order of the global numerical error of the test solver is a factor 50 smaller than the order of the numerical error of the train solver.

We also tested the proposed step adaptation algorithm on the Sphere2 dataset (see Figure 20), on MNIST (see Figure 19) and on CIFAR10 (see Figure 18). The behavior of the algorithm on different datasets as well as with different combinations of train and test solver supports the claims in the main text. It is interesting to note that if Midpoint is used as a train solver, the algorithm fluctuates around a much lower step size than if Euler is used as a train solver. A possible explanation for this behavior is that Midpoint is a second order Runge Kutta method which has therefore a lower numerical error than Euler which is a first order Runge Kutta method.

---

**Algorithm 2:** Step adaptation algorithm

---

initialize starting step_size $h$ according to (Hairer et al., 1993, p. 169);
**while** *Training* **do**
 batch = draw_batch(data);
 logits = model.do_forward_pass(batch, train_solver($h$));
 loss = model.calculate_loss(logits);
 train_solver_acc = model.calculate_acc(logits);
 **if** *Iteration % 50 == 0* **then**
  logits = model.do_forward_pass(batch, test_solver($h$));
  test_solver_acc = model.calculate_acc(logits);
  **if** *|train_solver_acc-test_solver_acc| > 0.1* **then**
   $h\_new$ = 0.5 $h$;
  **else**
   $h\_new$ = 1.1 $h$;
   **if** $h\_new > T$ **then**
    // Avoid increasing the step size indefinitely
    $h\_new$ = T;
   **end**
   logits = model.do_forward_pass(batch, train_solver($h\_new$));
   $h\_new$_acc = model.calculate_acc(logits);
   **if** *|test_solver_acc-h_new_acc| > 0.1* **then**
    $h\_new = h$;
   **end**
   $h = h\_new$
  **end**
 **end**
**end**

---

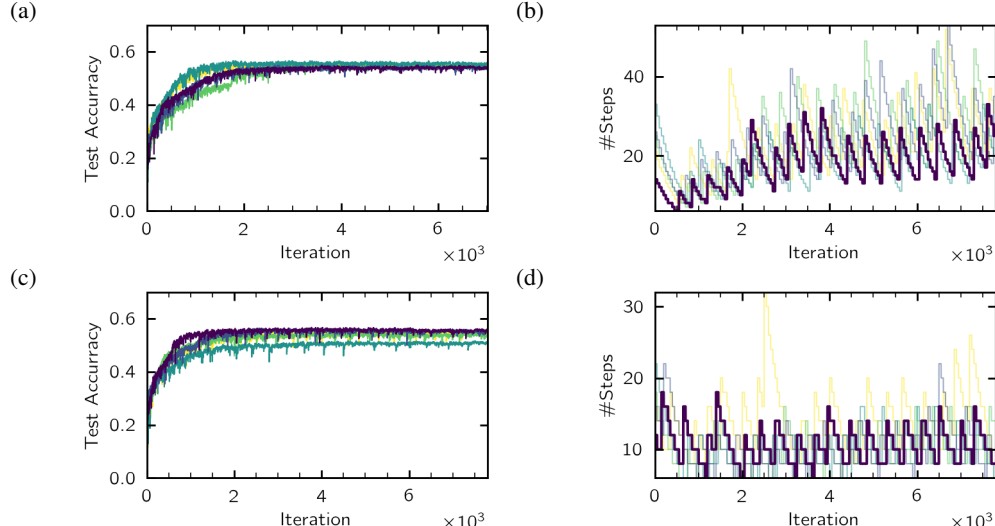

Figure 18: Using the step adaptation algorithm for training on CIFAR10. (a), (b) using Euler as train solver and rk4 as the test solver and (c), (d) using Midpoint as train solver and rk4 as the test solver. (a), (c) show the test accuracy over the course of training for five different seeds. (b), (d) show the number of steps chosen by the algorithm over the course of training.

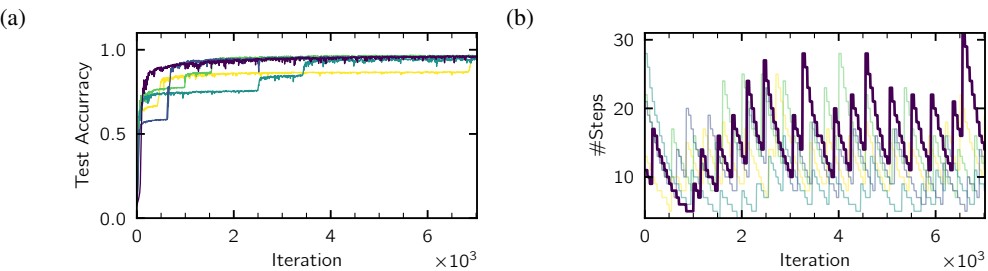

Figure 19: Using the step adaptation algorithm for training on MNIST. (a), (b) using Euler as the train solver and Midpoints as the test solver. (a) show the test accuracy over the course of training for five different seeds. (b) show the number of steps chosen by the algorithm over the course of training.

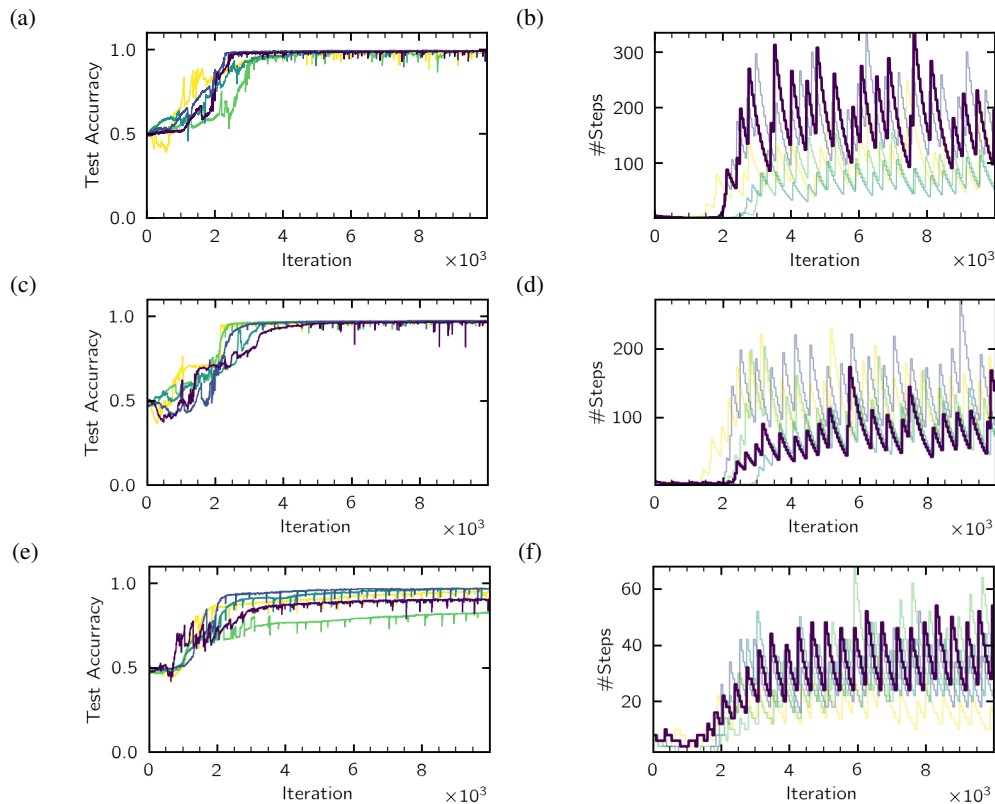

Figure 20: Using the step adaptation algorithm for training on Sphere2. (a), (b) using Euler as the train solver and Midpoints as the test solver (c), (d) using Euler as train solver and rk4 as the test solver and (e), (f) using Midpoint as train solver and rk4 as the test solver. (a), (c), (e) show the test accuracy over the course of training for five different seeds. (b), (d), (f) show the number of steps chosen by the algorithm over the course of training.

### C.2 TOLERANCE ADAPTATION ALGORITHM

The tolerance adaptation algorithm is similar to the step adaptation algorithm. In order to avoid infinitely increasing the tolerance, the algorithm checks whether the solver uses at least $min_{steps}$ which we set to 2 for our experiments. As for the fixed step solvers, the algorithm checks whether to accept the tolerance if the tolerance was increased.

For our experiments we used Fehlberg21 and Dopri54 as train solver and Dopri54 as test solver. To ensure that the test solver has a smaller numerical error than the train solver the test solver uses a smaller tolerance than the train solver. If Fehlberg21 is used as a train solver, than the tolerance for the test solver Dopri54 is set to 1/5 of the train solver tolerance. If Dopri54 is used as a train solver, than the tolerance for the test solver Dopri54 is set to 1/10 of the train solver tolerance.

The results for the tolerance adaptation algorithm are shown in Figure 21. After an initial adjustment phase, the number of function evaluations (nfe) and the tolerance fluctuate for the rest of training similar to the number of steps for the step adaptation algorithm. For different seeds, the tolerance fluctuates around different final values.

---

**Algorithm 3:** Tolerance adaptation algorithm

---

**Inputs** train_solver, test_solver, model;
initialize tolerance tol$= 10^{-6}$;
**while** *Training* **do**
    batch = draw_batch(data);
    logits = model.do_forward_pass(batch, train_solver(tol));
    loss = model.calculate_loss(logits);
    train_solver_acc = model.calculate_acc(logits);
    **if** *Iteration % 50 == 0* **then**
        logits = model.do_forward_pass(batch, test_solver(tol));
        test_solver_acc = model.calculate_acc(logits);
        **if** $|train\_solver\_acc\text{-}test\_solver\_acc| > 0.1$ **then**
            tol_new= 0.5 tol;
        **else**
            tol_new = 1.1 tol;
            **if** *steps $<= min\_steps$* **then**
                // Avoid increasing the tolerance indefinitely
                tol_new = tol;
            **else**
                logits = model.do_forward_pass(batch, train_solver(tol_new));
                tol_new_acc = model.calculate_acc(logits);
                **if** $|test\_solver\_acc\text{-}tol\_new\_acc| > 0.1$ **then**
                    tol_new = tol;
                **end**
            **end**
        **end**
        tol = tol_new
    **end**
    model.update(loss);
**end**

---

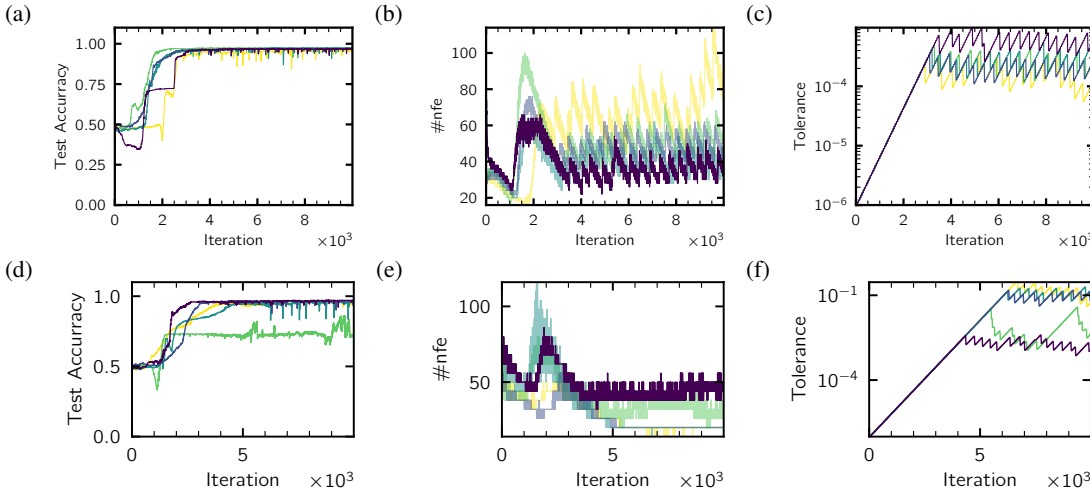

Figure 21: Using the tolerance adaptation algorithm for training on Sphere2 for Fehlberg21 as train solver, and Dopri54 as test solver (a), (b), (c). (d), (e), (f) Dopri54 was used as train and test solver where the test solver uses a reduced tolerance. (a), (d) shows the test accuracy over the course of training for five different seeds. (b), (e) shows the number of function evaluations the course of training. (c), (f) show the tolerance chosen by the algorithm over the course of training

# D  ARCHITECTURE AND HYPER-PARAMETERS

We chose the architecture for our network similar to the architecture proposed by Dupont et al. (2019). We tried to find hyperparameters which worked well for all step sizes. The same hyper-parameters were used for the grid search and for training with the step adaptation algorithm:

## D.1  ARCHITECTURE AND HYPER-PARAMETERS USED FOR MNIST

Neural ODE Block

- Conv2D(1, 96, Kernel 1x1, padding 0) + ReLu
- Conv2D(96, 96, Kernel 3x3, padding 1) + ReLu
- Conv2D(96, 1, Kernel 1x1, padding 0)

Classifier

- Flatten + LinearLayer(784,10) + SoftMax

Hyper-parameters

- Batch size: 256
- Optimizer: SGD (fixed step solvers), Adam (adaptive step size solvers)
- Learning rate: 1e-2 (fixed step solvers), 1e-4 (adaptive step size solvers)
- Iterations used for training: 7020

## D.2  ARCHITECTURE AND HYPER-PARAMETERS USED FOR CIFAR10

Neural ODE Block

- Conv2D(3, 128, Kernel 1x1, padding 0) + ReLu
- Conv2D(128, 128, Kernel 3x3, padding 1) + ReLu
- Conv2D(128, 3, Kernel 1x1, padding 0)

Classifier

- Flatten + LinearLayer(3072,10) + SoftMax

Hyper-parameters

- Batch size: 256
- Optimizer: Adam
- Learning rate: 1e-3
- Iterations used for training: 7800

## D.3  ARCHITECTURE USED FOR CONCENTRIC SPHERE 2D DATASET

Neural ODE Block

- Conv1D(1, 32, Kernel 1x1, padding 0) + ReLu
- Conv1D(32, 32, Kernel 3x3, padding 1) + ReLu
- Conv1D(32, 1, Kernel 1x1, padding 0)

Classifier

- Flatten + LinearLayer(2,2) + SoftMax

Hyper-parameters

- Batch size: 128
- Optimizer: Adam
- Learning rate: 1e-4
- Iterations used for training: 10000

## D.4 PYTHON PACKAGES

In our code we make use of the following packages: Matplotlib (Hunter, 2007), Numpy (Harris et al., 2020), Pytorch (Paszke et al., 2019) and Torchdiffeq (Chen et al., 2018).

# E    EXTENDED RESULTS

## E.1    CONCENTRIC SPHERE 2D

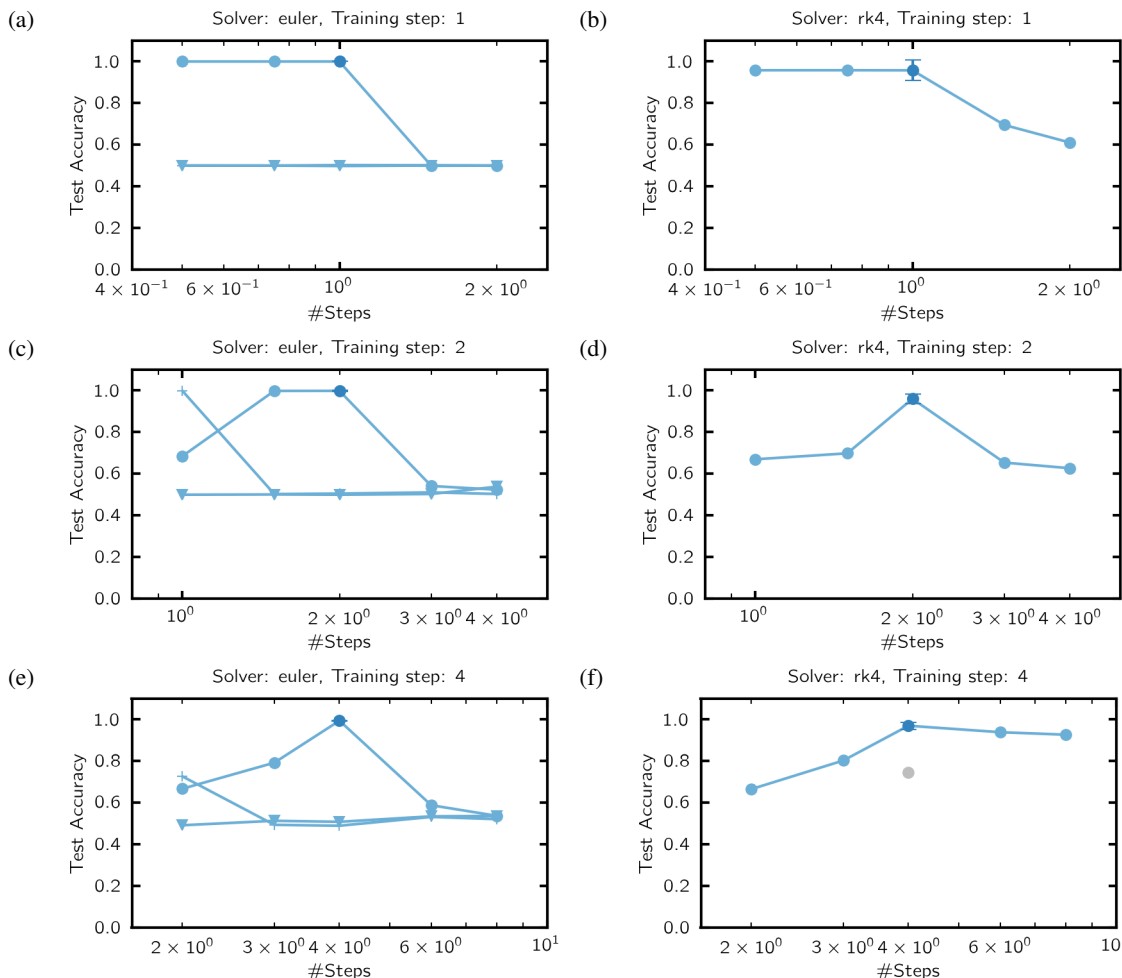

Figure 22: A Neural ODE was trained with different step sizes ((a), (b) 1 step, (c),(d) 2 steps, (e), (f) 4 steps) on the two dimensional concentric sphere dataset. The model was tested with different solvers and different step sizes. In (a), (c), (e) the model was trained using Euler's method. Results obtained by using the same solver for training and testing are marked by dark circles. Light data indicated different step sizes used for testing. Circles correspond to Euler's method, cross to the midpoint method and triangles to a 4th order Rung Kutta method. In (b), (d), (f) a 4th order Runge Kutta methods was used for training (dark circles) and testing (light circles). Excluded seeds are shown as grey circles.

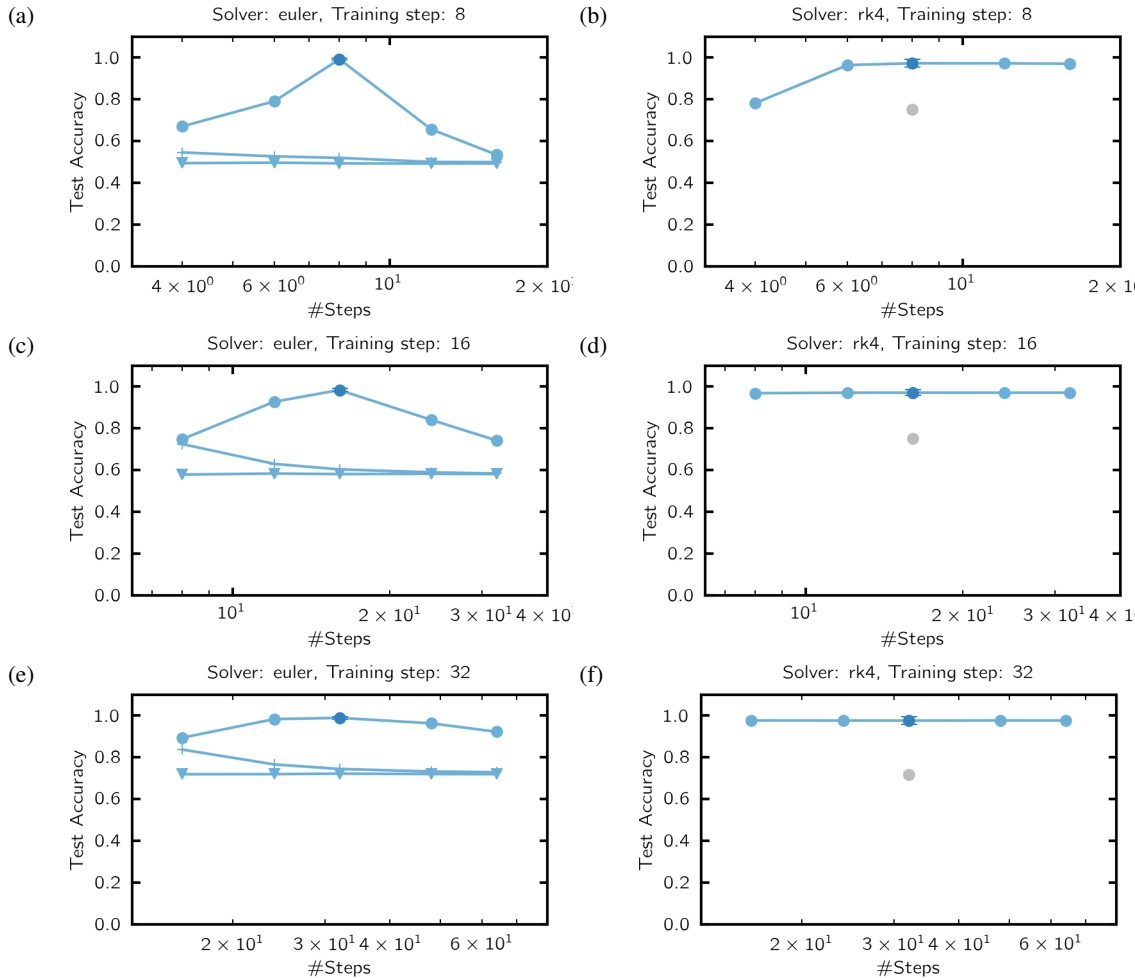

Figure 23: A Neural ODE was trained with different step sizes ((a), (b) 8 steps, (c),(d) 16 steps, (e), (f) 32 steps) on the two dimensional concentric sphere dataset. The model was tested with different solvers and different step sizes. In (a), (c), (e) the model was trained using Euler's method. Results obtained by using the same solver for training and testing are marked by dark circles. Light data indicated different step sizes used for testing. Circles correspond to Euler's method, cross to the midpoint method and triangles to a 4th order Rung Kutta method. In (b), (d), (f) a 4th order Runge Kutta methods was used for training (dark circles) and testing (light circles). Excluded seeds are shown as grey circles.

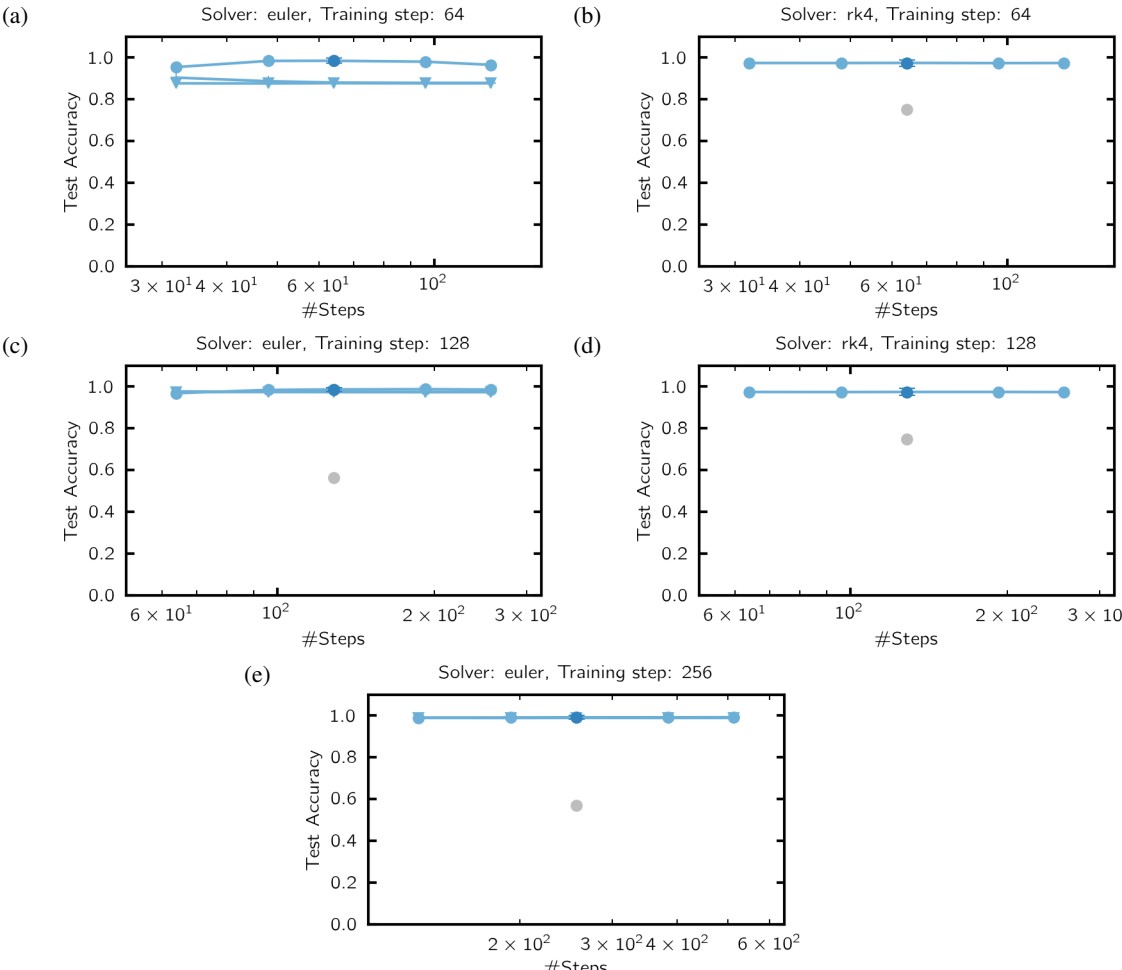

Figure 24: A Neural ODE was trained with different step sizes ((a), (b) 64 steps, (c),(d) 128 steps, (e) 256 steps) on the two dimensional concentric sphere dataset. The model was tested with different solvers and different step sizes. In (a), (c), (e) the model was trained using Euler's method. Results obtained by using the same solver for training and testing are marked by dark circles. Light data indicated different step sizes used for testing. Circles correspond to Euler's method, cross to the midpoint method and triangles to a 4th order Rung Kutta method. In (b), (d) a 4th order Runge Kutta methods was used for training (dark circles) and testing (light circles). Excluded seeds are shown as grey circles.

## E.2 CIFAR10

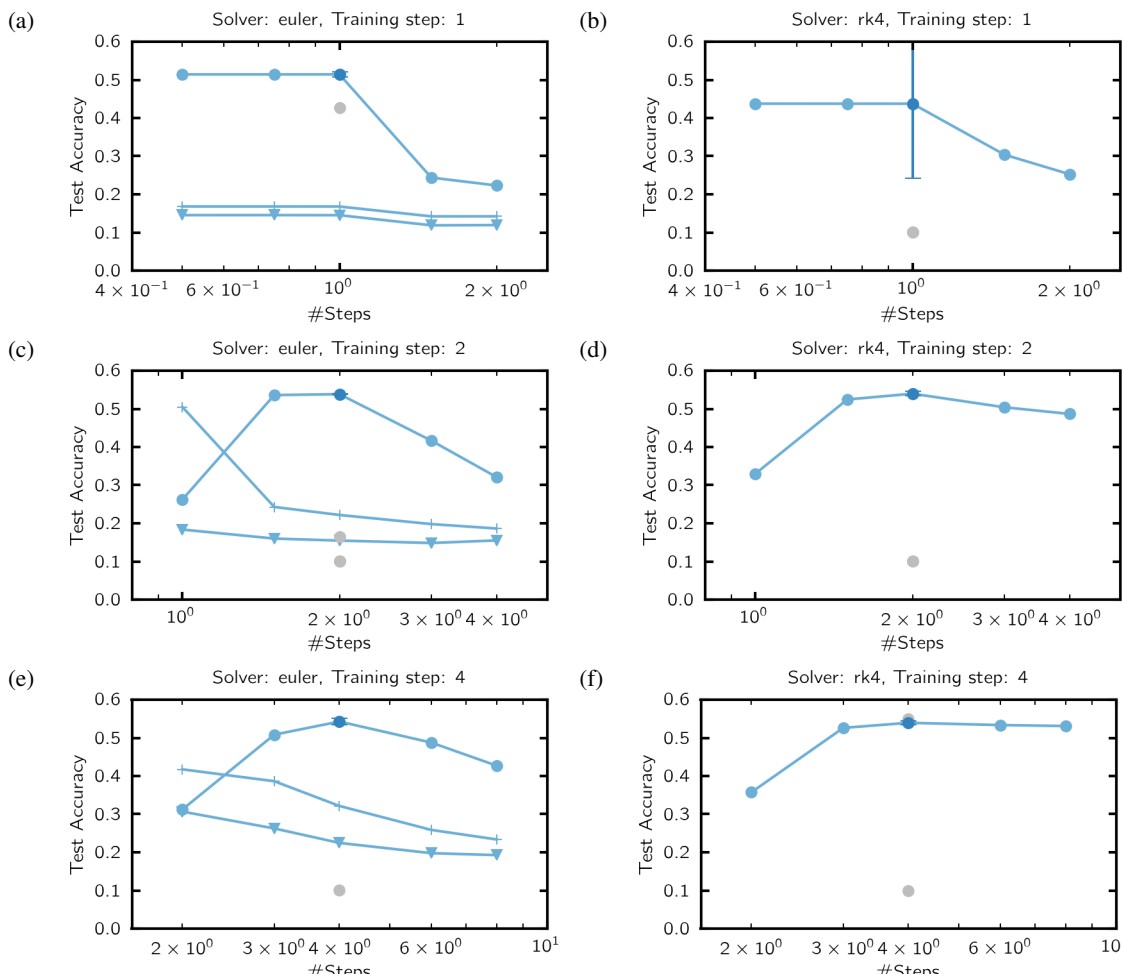

Figure 25: A Neural ODE was trained with different step sizes ((a), (b) 1 step, (c),(d) 2 steps, (e), (f) 4 steps) on the cifar10 dataset. The model was tested with different solvers and different step sizes. In (a), (c), (e) the model was trained using Euler's method. Results obtained by using the same solver for training and testing are marked by dark circles. Light data indicated different step sizes used for testing. Circles correspond to Euler's method, cross to the midpoint method and triangles to a 4th order Rung Kutta method. In (b), (d), (f) a 4th order Runge Kutta methods was used for training (dark circles) and testing (light circles). Excluded seeds are shown as grey circles.

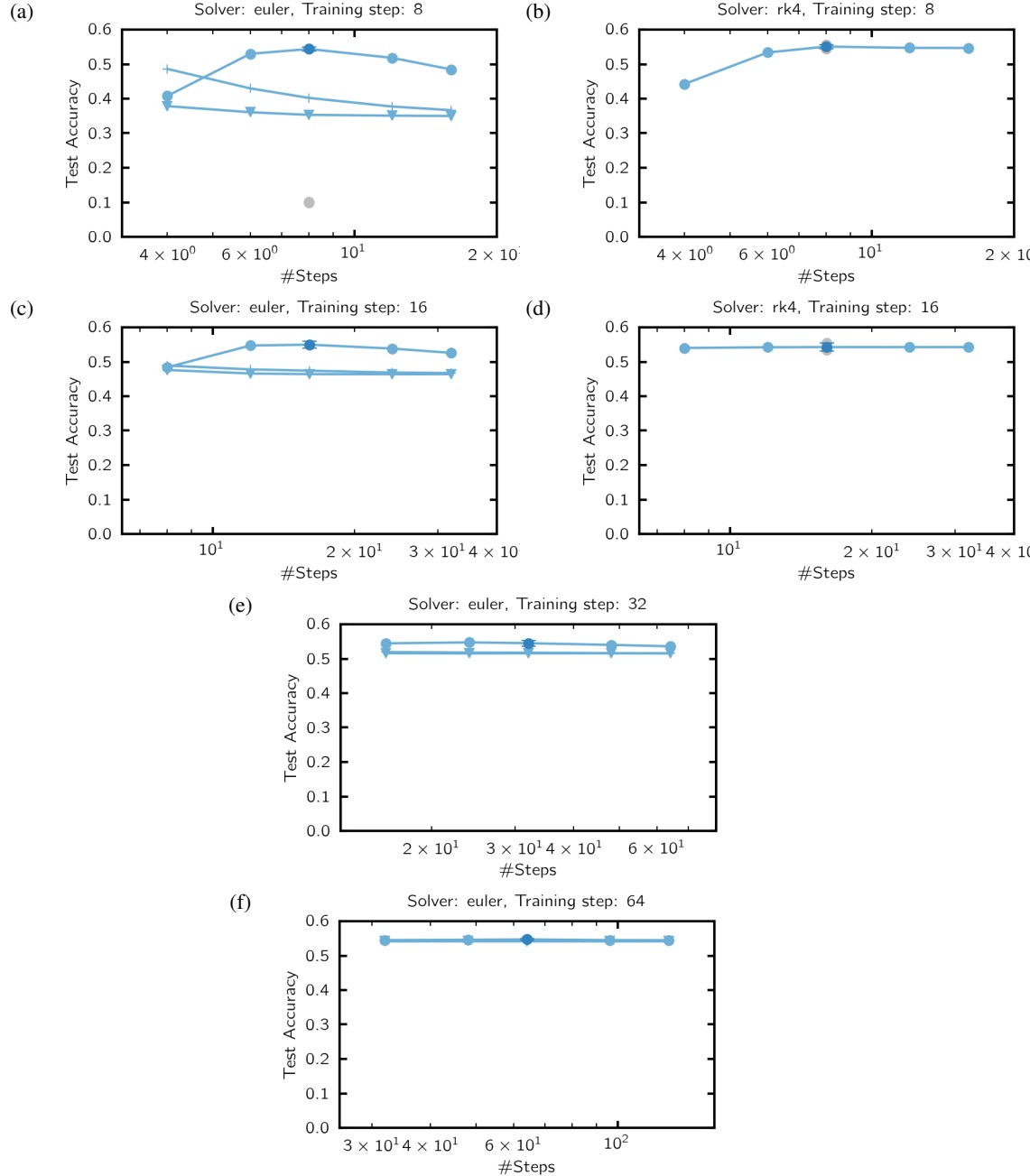

Figure 26: A Neural ODE was trained with different step sizes ((a), (b) 8 step, (c),(d) 16 steps, (e) 32 steps, (f) 64 steps) on the cifar10 dataset. The model was tested with different solvers and different step sizes. In (a), (c), (e), (f) the model was trained using Euler's method. Results obtained by using the same solver for training and testing are marked by dark circles. Light data indicated different step sizes used for testing. Circles correspond to Euler's method, cross to the midpoint method and triangles to a 4th order Rung Kutta method. In (b), (d) a 4th order Runge Kutta methods was used for training (dark circles) and testing (light circles). Excluded seeds are shown as grey circles.

