# OpenReview forum: "ResNet After All: Neural ODEs and Their Numerical Solution"
_ICLR.cc/2021/Conference — ICLR 2021 Poster_

### Official Review · AnonReviewer1 · 2020-10-28
**An Investigation of the discrete dynamics of Neural ODEs**

**Rating:** 6
**Confidence:** 4

**Review:**

**Summary**
The authors show that Neural ODEs exploit the ODE-solver used for training to realize a dynamical system that violates the ODE vector field property of non-overlapping trajectories. The authors conclude that NODEs are not real ODEs, hence the paper's title "ResNet after all.". To avoid such behavior, the authors propose to monitor the accuracy metrics using a finer ODE solver and decrease the solver's step size if a discrepancy between the two different stepsize accuracies is observed.

**comments**
While this paper's claims and experiments are relatively narrowly focused, the overall conclusion and proposed solution are clear.
However, I see a fundamental issue with the assumption of fixed stepsize solvers. The main reason why using adaptive stepsize solvers is to avoid such a problem of choosing the right stepsize. Consequently, I expect the described problem to be more elegantly
solved using a lower relative tolerance value of a dynamic stepsize solver.

In algorithm 2, there is a variable called test_acc. The term "test_acc" is overloaded in this context, and it can refer to the test-set accuracy or the training accuracy under a higher-order ODE solver.  If the authors refer to the training accuracy under a higher-order ODE solver (which I assume), please change the variable's name.

---

> ### Author Response · Authors · 2020-11-11
> **Re: Reviewer 1**
>
> We thank the reviewer for their comments.
>
> 1. **Adaptive step methods solve the problem:** We would like to thank the reviewer for this comment. We would like to point out that our work considers both adaptive and fixed step solvers (see Figure 4 (c) and (d) where adaptive step size methods where used for training the model.) Adaptive step size methods do not solve the problem described in the paper. Adaptive step size methods control *the local error* with a control signal typically specifying a desired number of significant digits. In this setting, the user has already done two things: run a preliminary analysis how many digits are required to likely get a qualitatively correct numerical solution and b) understood how much accuracy is required in the solution. In Neural ODEs, the required numerical accuracy depends on the downstream layers that change per gradient step. That is, the global error prerequisite of the numerical solver potentially changes with every gradient step. While there always exists a low enough tolerance such that the adaptation issue does not occur, this low enough tolerance may be prohibitively small in practice and is certainly leaving runtime efficiency on the table. We will try to make this point clearer in the paper.
> 2. We thank the reviewer for this suggestions and we will change the variable name to improve clarity.

---

### Official Review · AnonReviewer3 · 2020-10-28
**Good contribution but algorithm is not sound**

**Rating:** 7
**Confidence:** 4

**Review:**

- Paper makes a good contribution by pointing an intrinsic flaw in the NeuralODE technique. The problem is that even with an  error accruing step size, the results of a NerualODE can be good, leading to a false belief that the ODE used in the construct represents the phenomena, but instead it is the dynamic behaviour arising from the mixture of the ODE and the solver that separates the classes well.
- However, the proposed solution does not seem convincing. It seems like a work in progress. The solution is proposed in algorithm 2,4,6 which should have been algorithm 1,2,3.
- It is not made clear how solvers would be able to use this algorithm.
-The  algorithm is not nicely constructed.  Putting a function like "calculate accuracy higher order solver();" in an algorithm without fully describing what it does , is not advised.
- Figures are not illustrative, there is too much clutter. I believe a point could be made with the same amount of figures but with less clutter.
- Based upon the contribution made by the authors, it seems appropriate that their results are published right now.

---

> ### Author Response · Authors · 2020-11-11
> **Re: Reviewer 3**
>
> We thank the reviewer for their comments.
>
> 1. **Maturity of proposed solution** We agree that the proposed solution is a first simple heuristic to approach the problem. It may not provide overwhelming success, but it does solve the problem. We hope that this helps the community to understand what the intrinsic flaw is all about and that it is solvable in general, thus leading to further research on this problem.
>
>    We would like to understand in more detail why you did not find the algorithm convincing such that we can improve our work or add some discussion. We will fix the latex bug concerning the numbering of the algorithm - thank you for spotting this!
>
> 2. We will adapt the algorithm such that it represents the full training loop.
> We would like to point out that the presented algorithm is not part of the solver but outside of
> the solver and can thus be applied to any solver.
> We would like to ask the reviewer to point out where we should provide additional clarification.
>
> 3. The presentation of the adaption algorithm can indeed be improved. We will add additional details on the `calculate_accuracy_higher_order_solver()` function in the text and/or the appendix. We will also work on improving the overall clarity of the algorithm.
>
> 4. We promise to improve the visualization on the experiments. We will do this by incorporating the suggestions by reviewer 4 minor comment 2. We are open to concrete suggestions on how to reduce the clutter in our figures.
>
> We hope to have a first revision including the above changes by the weekend.
> We are currently running additional experiments and we will only update the figures
> after we have collected all results. This revision will take a bit longer.

---

### Official Review · AnonReviewer4 · 2020-10-28
**Review for Neural ODEs and Their Numerical Solution**

**Rating:** 7
**Confidence:** 3

**Review:**

This paper empirically studies whether Neural ODEs have a valid ODE interpretation. The authors show that a Neural ODE model does not necessarily represent a continuous dynamical system if the discretization of the numerical method is too coarse. Indeed, this is a widely overlooked issue that has been largely ignored in the Neural ODE community. To address this issue, the authors propose a novel adaptive step size scheme.

Reasons for my score: Overall, I vote for marginally above acceptance threshold. The presented ideas and results are very interesting and relevant for the community. It is important to better understand if and when a Neural ODE has a valid ODE interpretation. I am willing to increase my score if the authors can address my concerns during the rebuttal period.

Pros:
-------
+ The work addresses a crucial aspect of Neural ODEs that is particularly important in the context of scientific and robotics applications. That is, because here one is often not only interested in the predictive accuracy of the model, but also whether the model has a valid ODE interpretation.

+ The authors show several illustrating examples that demonstrate how the Neural ODE is affected by the specific solver configuration used for training.

+ The adaptive step size scheme is simple yet effective. The experiments clearly demonstrate the advantage of this scheme.


Cons:
-------

- It would be good if you apply your adaptive step size scheme to an actual scientific problem, where it actually matters that the ODE has a valid ODE interpretation. It is not intuitive why an ODE interpretation is relevant for computer vision tasks.

- An extended set of results in Section 3.1 would help to better understand the performance of the proposed algorithm. For instance, how does the results change if you use RK4 for testing, instead of midpoint (this should only be marginally more expensive). Also, what happens if you train with midpoint and then use RK4 for testing. Next, can you also add results for MNIST or FMNIST in Table 1 in order provide an additional set of experiments.

- The Neural ODE block that you consider is very shallow. I assume, that it should be possible to achieve about 75% accuracy on CIFAR10 using a state of the art Neural ODE block.

- The Appendix is poorly formatted. Typically, I would expect that Figures are embedded into a descriptive text. It would be nice if you provide at least some discussion for why you provide these Figures and what we can learn from it (in addition to the captions). Also, it is not clear to me why you are presenting a `'preliminary tolerance adpation algorithm'. Are you proposing to use this algorithm, or is this an idea for a future work that still needs to be tested and improved? Typically, I would not expect to see any preliminary results in a conference paper.

- The authors miss to discuss how their works relates to some recent theoretical results [1,2,3].

- The title of the paper seems not fitting, i.e., what do mean by `ResNet after all'? You do not discuss ResNets in much detail in this paper.

- Please provide code in order to reproduce the results.

Minor comments:
-------

* Some parts of the paper are unclear. For instance, the authors do not establish the relationship between step size and number of steps. This should be discussed somewhere below Eq. (3). On page 7 you say `that 'after a pre defined number of steps (we chose k= 50)'. I assume here you refer to the number of iterations?

* Some of the Figures are crowded and difficult to parse. For instance, there is much going on in Figure 4. First, it would help if you increase the width of the plots (there is lot's of white space on the left and right of the figure). Further, it would help if you reduce the content slightly. Finally, a legend would be very much appreciated.

* I think, it sounds better to use 'adaptation' instead of 'adaption'.


References
-----

[1] Bo, Lijun, Agostino Capponi, and Huafu Liao. "Deep Residual Learning via Large Sample Mean-Field Optimization: Relaxed Control and Gamma-Convergence." arXiv:1906.08894 (2019).
[2] Thorpe, Matthew, and Yves van Gennip. "Deep limits of residual neural networks." arXiv preprint arXiv:1810.11741 (2018).
[3] W. E, J. Han, and Q. Li, "A mean-field optimal control formulation of deep learning," Research in the Mathematical Sciences, vol. 6, no. 1, p. 10, 2019.

---

> ### Author Response · Authors · 2020-11-11
> **Re: Reviewer 4**
>
> We thank the reviewer for their helpful comments.
>
> 1. **Application to scientific problem:** We agree that it is currently unclear when a valid ODE interpretation would be required and that it might not be relevant to computer vision tasks yet. We believe that the Neural ODE model class---as opposed to ResNets---is only beginning to develop its full potential. For instance, [1] discusses that Neural ODEs as a model class might improve robustness. Therefore, we believe our work to be relevant beyond the immediate justification.
>
>  That being said, we can try to run experiments, if you have a concrete application in mind?
>
> 2. **Extended results:** The suggestions by the reviewer for extending the experiments with the step adaption algorithm are interesting. We agree that the step adaption algorithm should also work for the proposed cases.If time permits we will run additional experiments with the step adaption algorithm. We will start additional experiments for mnist and add them to the paper as soon as they are finished. We cannot promise these results until the end of the rebuttal but we hope that everything finishes in time.
> 3. Please see  our answer to Reviewer 2 point 5.. Additionally, to improve the performance of a "single"  ODE-block one could add additional augmented dimensions [2] (with the same number of parameters they achieve 60.6% on CIFAR-10).
> 4. We will work on reformatting the appendix and add additional explanatory text. "preliminary algorithm": We have called this algorithm preliminary as the results have been below our expectations compared to the relative success in the fixed step case. The problem of continuous vs. discrete semantics also appears for adaptive methods but the simple heuristic algorithm we present does not seem to work as well as expected for adaptive methods. We want to include the results for the tolerance adaptions algorithm nonetheless as we think they emphasize our overall findings. We are aware that the presented algorithm is only the first step of many in the direction of solving this problem and we hope that the community will continue to tackle this challenge. We will remove the word preliminary as it is unfitting as pointed out by the reviewer.
> 5. We thank the author for point us towards these publications and we will add these papers to our discussion.
> 6. We would like to point out that Reviewer 1 seems to find the title quite fitting. Nevertheless, we are open to suggestions. One option would be the title we chose for a previous version of this paper: "When are Neural ODEs proper ODEs?"
> 7. We are currently applying for clearance of the code from our institution. We hope to achieve this before the end of the discussion phase, but we promise to release the code eventually.
>
> ## Minor comments
> 1. We thank the reviewer for pointing out this detail. We will add an explanation for the relationship between step size and number of steps in the introduction. We will also fix the wording on page 7 (number of steps -> number of iterations).
> 2. We will incorporate the helpful suggestions of the reviewer. We will increase the width of the figures and we will also add a legend to make the figures clearer.
> 3. We thank the reviewer for this suggestion and we will adapt the paper to use the word"adaptation".
>
> We hope to have a first revision including all points of clarity by the weekend. A revision including improved figures and, resources permitting, experiments will take a bit longer.
>
> ## References
> [1] On Robustness of Neural Ordinary Differential Equations, Yan et al, 2020
>
> [2] Augemented Neural ODEs, Dupont, Doucet, Teh, 2020

---

> > ### Comment · AnonReviewer2 · 2020-11-18
> > **Original title is good**
> >
> > re: title suggestions
> > I think "ResNet After All: Neural ODEs and Their Numerical Solution" is a very apt title for this paper!

---

> > > ### Comment · AnonReviewer4 · 2020-11-18
> > > **I am okay with the title**
> > >
> > > To be fair, I have no hard feelings about the title. My point is that the title seems potentially misleading. A Neural ODE might not necessarily represent a continuous dynamical system if the discretization of the numerical method is too coarse, and in this case a Neural ODE is a ResNet after all! However, if the Neural ODE model is carefully trained then it does learn an approximation for the underlying continuous dynamical system, and in this case it is not a ResNet. That is an important aspect and it should be better discussed somewhere in the abstract, introduction or in the conclusion.

---

> > > > ### Comment · AnonReviewer2 · 2020-11-18
> > > > **Minor title change**
> > > >
> > > > I see your point. What about "ResNet After All? Neural ODEs and Their Numerical Solution" (colon replaced by question mark)?

---

### Official Review · AnonReviewer2 · 2020-10-28
**Review of "RESNET AFTER ALL: NEURAL ODES AND THEIR NUMERICAL SOLUTION"**

**Rating:** 5
**Confidence:** 4

**Review:**

Paper summary:

The paper demonstrates how neural ODE models generating features for downstream tasks (or simply modelling trajectories) may rely on the discreteness of integration methods to generate features and thus fail in the exact ODE limit of integration step-size going to zero. The paper highlights particular failure modes, such as the discreteness of integration methods allowing for qualitative differences like overlapping trajectories (impossible for the exact solution of an autonomous ODE) compared to exact solutions, or quantitative differences like the accumulated error of a numerically integrated ODE resulting in useful features for downstream tasks. The paper empirically demonstrates the phenomenon that low training losses can be achieved for a range of integration methods and integration step-sizes, but that, of these models, the ones robust to changes in integration method and decreases in integration step-sizes at test time are those trained below a certain (empirically determined) integration step-size threshold. This is attributable to models trained with lower integration step-sizes maintaining features that are qualitatively the same as or quantitatively close to those features produced by the same model with smaller integration step-sizes. The paper proposes an algorithm for adapting integration step-size during training so that the resulting neural ODE model is robust to changes in integration method and integration step-size at test time. The algorithm is empirically demonstrated to achieve the same performance as grid search (for similar numbers of function evaluations).

------------------------------------------
Strengths and weaknesses:

I liked the paper as it raised an important question of whether and when we should interpret neural ODEs as having continuous semantics and gave a few examples of failure cases. The results of the step-size adaptive algorithm were also promising (it matched grid search but with less work). Further, the paper was clearly written and easy to understand.

However, as it stands, I’m assigning a score of 5. I like the paper and think that it would be a good workshop paper but is not ready for the main conference. The reason for this is that the theoretical part of the paper is mostly qualitative, whilst the experiments are not extensive enough to make up for the qualitative theoretical justification. If one of these two areas were to be improved, I would be happy to increase my score. To be concrete, here are examples of theoretical and empirical questions whose answers (just one would do) would increase the paper’s score for me:

1)	How can we mathematically describe when numerically integrated trajectories cross over in terms of the time over which the ODE is integrated and on the initial separation of the trajectories?

2)	Suppose we are integrating an ODE for which we have the analytic form. Are there additional behaviours we need to watch out for? For example, after passing below a step-size where we transition from crossing trajectories to non-crossing trajectories, is it possible to transition back to crossing trajectories as we continue to decrease the step-size? Or can we rule out this case, for example, in the case of a f being continuous in the equation z’(t) = f(z)?

3)	For Lady Windermere’s Fan with the true dynamics, at what step-size does trajectory overlap cease to occur (assuming a minimum initial separation of trajectories and fixed time period)? And if we instead attempt to learn Lady Windermere’s Fan with a neural ODE, at what step-size does the neural ODE start to be robust against test-time decreases in step-size? How does this latter step-size compare to the former step-size?
------------------------------------------
Questions and clarification requests:

1)	What was the true underlying model for figures 1 and 2?

2)	Why are the classifier decision boundaries different in figures 2a and 2b? I thought that you trained a neural ODE with h_train = 1/2 and then tested this model for both h_test = 1/2 and h_test = 1/4.

3)	I didn’t understand the connection between Lady Windermere’s Fan and the XOR problem. Does running Lady Windermere’s Fan on R^2 with an XOR labelling lead to trajectory end points that are linearly separable? If so, how did you discover this?

4)	You mention at the end of section 2.2 that “The current implementation of Neural ODEs does not ensure that the model is driven towards continuous semantics as there are no checks in the gradient update ensuring that the model remains a valid ODE nor are there penalties in the loss function if the Neural ODE model becomes tied to a specific numerical configuration.” Do you have any ideas of what directions you might head in in terms of regularising neural ODEs so that they manage to learn continuous semantics, even when trained at larger step-sizes? In particular, why did you go in the direction of an adaptive optimization algorithm, instead of, say, training the neural ODE with a randomly chosen step-size each iteration or even step?

5)	Why was the CIFAR-10 classification accuracy (~55%)? Previous work on neural ODEs has obtained accuracy in the 80 -95% range. Is this just due to the limited expressiveness of the upstream classifier, cf. “For all our experiments, we do not use an upstream block f_u similar to the architectures proposed in Dupont et al. (2019). We chose such an architectural scheme to maximize the modeling contributions of the ODE block.”

------------------------------------------
Typos and minor edits:

- Write Initial Value Problem (IVP) on first usage of IVP.
- Fig.2 caption – “The model was trained …, we used …” -> “The model was trained …, and we used …”
- Page 8, Conclusion, line 3 – “… an continuous…” -> “… a continuous …”

---

> ### Author Response · Authors · 2020-11-11
> **Re: Reviewer 2 Theoretical and empirical questions**
>
> We would like to thank the reviewer for their detailed comments.
>
> 1. **Number of crossings:** We thank the reviewer for this interesting question. We will get back to you if we find a solution.
>
> 2. **Interplay of solver and vector field** We would like to work on answering this interesting question and therefore we ask the reviewer to please clarify a few details. Particularly, we kindly ask the reviewer to clarify what *analytical form of an ODE* is referring to. We believe that reviewer is either referring to that the analytical form of the right side of ODE f(z) is known or that the analytical solution to the ODE is known. We cannot think of a scenario where one would use a numerical solver for an ODE with known analytical solution, so we currently assume the reviewer considers the case of a right-hand side with known analytic properties.
>
>    What can be said at this point: Finding a step size h* for which no crossings occur does not guarantee that no crossings will occur for all step sizes below this step size h*. However, for each problem, there exists some sufficiently small step size h_s such that for all h < h_s, no crossings do occur. I.e., an empirical first occurence of no crossings is not a sufficient condition for small enough step size, but a small enough step size always exists for which even smaller step sizes will produce no crossings.
>
>    Continuous right-hand sides are also not sufficient to eliminate the problem of trajectory crossings. We can construct a synthetic corner case highlighting these problems which we will add in the appendix  (space permitting in the main text).
>
> 3. **Lady Windemere's Fan:** We will provide additional explanations concerning Lady W.'s fan see also our answer
> to question 3 below. We would like to know whether your question refers to the specific ODE presented in Eq. (4) or problems
> where Lady W.'s fan can be observed in general? Note that Lady W.s fan is a problem *independent* of the trajectories crossing problem. In this case, valid ODE semantics are maintained, but the trained model still crucially depends on the discrete solver dynamics. We will try to make this clearer in the main text.
>
> We hope to have a first revision including the above changes by the weekend.
> The addition of the synthetic corner case might take a bit longer, though.

---

> > ### Comment · AnonReviewer2 · 2020-11-18
> > **Clarifying reviewer 2 theoretical and empirical questions**
> >
> > re: 2. Interplay of solver and vector field
> >
> > My apologies for the confusion about the use of “analytic”. I wanted to suggest a concrete question for the authors to answer, and so I wanted to suggest answering the questions in the rest of the paragraph for the case of f(z) being an analytic function (I could equally have said some other condition like infinitely differentiable, continuously differentiable, continuous, and so on). And, unfortunately, I confused the matter by talking about f(z) being continuous at the end of the paragraph. What I was intending to say was that, assuming some properties of f(z) that you should feel free to choose, can you give any mathematical guarantees on the presence/absence of crossing over?
> >
> > The points you mention are exactly the sort of questions I think the paper would benefit from exploring more. In particular, you write:
> > 1) “Finding a step size h* for which no crossings occur does not guarantee that no crossings will occur for all step sizes below this step size h*.”
> > 2) “However, for each problem, there exists some sufficiently small step size h_s such that for all h < h_s, no crossings do occur.”
> > 3) “I.e., an empirical first occurence of no crossings is not a sufficient condition for small enough step size, but a small enough step size always exists for which even smaller step sizes will produce no crossings.”
> >
> > For example, going down the theoretical route, are there any theorems proving that “a small enough step size always exists for which even smaller step sizes will produce no crossings”? And what conditions are required for such theorems to hold? Presumably these conditions will require a mathematical definition of “crossing over” (I know you’re looking into this re: 1. Number of crossings).
> >
> > Meanwhile, for a given ODE, can you always find “a step size h* for which no crossings occur” that “does not guarantee that no crossings will occur for all step sizes below this step size h*”? How many times can a numerically integrated ODE alternate between non-crossing and crossing as step-size is decreased?
> >
> > -----------------
> > re: 3. Lady Windemere's Fan
> >
> > I realise after reading your comments that I'd misunderstood what Lady Windemere's Fan was (see my reply to your other comment). What I meant with my question was this:
> > -	Suppose we have an ODE z’(t) = f(z). We could take equation (4) if we liked, but any ODE will do.
> > -	Suppose we integrate it with a given method. At what threshold step-size does trajectory overlap cease to occur (and for all smaller step-sizes)?
> > -	Suppose we train a neural ODE (and use the same method as above for the integration) to predict z(T) from z(0) for some specific time T. At what threshold step-size does test accuracy with step-sizes below this threshold cease to fall? E.g. we obtain 100% train accuracy when training on any step-size, but only with training step-size 10^{-3} does test accuracy with step-sizes less than 10^{-3} continue to be near 100%.
> > -	How do these two threshold step-sizes compare?

---

> > > ### Author Response · Authors · 2020-11-19
> > > **Re: Clarifying reviewer 2 theoretical and empirical questions**
> > >
> > > Thank you for your interest in the problem and all the interesting questions.
> > >
> > > We have added a new section to the appendix including new analysis plots and a new plot for Lady W.s fan.
> > >
> > > Both trajectory crossing and Lady W.'s fan contribute to the global error.
> > > Heuristically, there exists a step size where the sensitivity of the global error
> > > is smaller than the sensitivity of the downstream layer.
> > > Therefore, there should exist an h* such that for all h<h* the accuracy no longer changes.
> > > We discuss this at the end of Section 2.2 but maybe to emphasize this point more.
> > > Overall, we are unaware of any theoretical statement supporting our heuristic statements.
> > > The difficulty we see is that one has to not only look at the theory of an individual IVP but of
> > > several IVPs.
> > > We have looked into this and are not aware of any theoretical work in this area.
> > > Overall there are many subtle effects and not every model violation shows in the test accuracy.
> > > For example if we have inter-class crossings between trajectories, this does not change the
> > > test accuracy.
> > >
> > >
> > > Does this answer some of your questions? Are there any specific points you would like us to clarify?
> > > Additionally, are there any effects we should provide more detail on in our paper?

---

> > > ### Author Response · Authors · 2020-11-23
> > > **End of revision period fast approaching**
> > >
> > > Unfortunately, the revision period is ending soon. Reviewer 2, are there any last concerns/questions that you want us to address? We might not be able to improve the manuscript further until revision deadline, but we could try to incorporate them into a potential camera-ready version.
> > >
> > > Furthermore, have we addressed your concerns adequately to maybe improve your judgement of our paper?

---

> ### Author Response · Authors · 2020-11-11
> **Re: Reviewer 2 Question and Clarification Requests**
>
> ## Questions and Clarification requests
> 1. We thank the reviewer for this question. For Figure 1 and 2, there are no true underlying dynamics.The Neural ODE model is only given the classification task and has to find some dynamics which solve the problem. In combination with the classifier, many different vector fields might be possible. We will clarify this in the main text.
>
> 2. We thank the reviewer for pointing this out. The difference in the images is due to different scaling of the axis which were chosen such that the final position of all points is shown. We will add units to the axis of Figure 2 (a) and (b) to make this clearer.
>
> 3. Our descriptions of Lady W.'s fan indeed needs further explanation - we will expand section 2.2. Lady W.'s fan does not refer to a specific problem, but how the local error gets accumulated into the global error. Lady Windemere's fan does not guarantee solving the XOR problem. But the example used in Section 2.2 is aimed towards showing that error accumulation can lead to dynamics which solve the XOR problem, even if the analytic solution to the ODE does not. The ODE we present in section 2.2. corresponds to a flow with increasing ellipsoids and an increase in the rotational speed for this problem. We discovered this model based on the knowledge that the precision of the solver influences how the rotational speed of the ellipsoids is resolved.
>
> 4. "Do you have any ideas of what directions you might head in in terms of regularising neural ODEs so that they manage to learn continuous semantics, even when trained at larger step-sizes?" - We thank the reviewer for this interesting question. We do not have any precise ideas yet but restricting the Lipschitz constant of the Neural ODE to below 1 avoids crossing trajectories [1]. Additionally, forcing the model to learn simpler dynamics could reduce the critical step size (as done for example in [2]). We will add this discussion to the main text of the paper.
> "In particular, why did you go in the direction of an adaptive optimization algorithm, instead of, say, training the neural ODE with a randomly chosen step-size each iteration or even step?" The idea of our algorithm is to keep the number of steps as small as possible. The idea of reviewer to use random step sizes is interesting. Random step sizes might not provide the wanted gradient information, as too large step sizes drive the system to discrete dynamics. Therefore, if the variance in the steps is too large, we believe that training might become difficult.
>
>  5. **Low accuracy on cifar:**  The performance of our model is due to the simple architecture chosen for our experiments. To improve performance, other work uses a deeper classifier, an upstream downsampling block and often even multiple ODE blocks. We did  not want to use an upstream block, a deeper downstream classifier block and multiple ODE blocks, as we want to maximize the contribution of the ODE block. We will add this to the explanation in the main text
>
>  ## Typos and minor edits
> Thank you for spotting these and we will fix them.
>
>  ## References
>  [1] Invertible Residual Networks, Behrmann et al., ICML, 2019
>
>  [2] Learning differential equations that are easy to solve, Kelly et al., arXiv, 2020

---

> > ### Comment · AnonReviewer2 · 2020-11-17
> > **Thanks for helpful clarifications**
> >
> > Thanks for clarifying the points I was unsure on. The edits you've made in the updated version have cleared up the all questions I had.
> >
> > re: Lady Windemere's fan.
> > I think the sentence "However, in Figure 3 (a), the accumulation of error in the numerical solution (coined as Lady Windermere’s Fan in Hairer et al. (1993, x 1.7)) results in a valid feature for a linear decision (classification) layer" threw me off. I mistakenly thought Lady Windermere's Fan was the numerical solution to equation 4, as opposed to the phenomenon of accumulation of error in the numerical solution to an ODE.

---

### Author Response · Authors · 2020-11-11
**Re: all**

We thank the reviewers for their detailed and insightful comments. Reviewers 2, 3 and 4 seem to agree that our paper presents an important issue with high relevance to the Neural ODE community. Reviewers 2 and 4 raise concerns regarding the experimental coverage and the theoretical underpinning, leading to hesitation whether this submission should be accepted now or at a future conference. Reviewer 1 is skeptical about the generality of the presented problem.

We will try to address the reviewers' concerns within the rebuttal period and we will try to convince you that the work is indeed ready to be published now. To this end, we will improve the clarity and presentation and we will also try to add missing experiments.

For individual feedback, please refer to the comments below your respective reviews.

---

### Author Response · Authors · 2020-11-13
**Revision 1**

We have incorporated first improvements and uploaded a new draft of our paper.

These improvements include:

- We worked on improving the clarity of the text in Section 2 specifically with regard
to Lady W.'s fan. We have also added ticks and tick labels to the axis of Figure 2.
Reviewer 2 does this help you in regard to your Question 3? Additionally, we will try
to improve this Section further by introducing additional figures.
- We have improved the algorithm by using more precise notation
 and we hope this makes it clearer how the proposed algorithm can
be used in practice as suggested by Reviewer 3 and Reviewer 1.
- We have added the references suggested by Reviewer 3 to our related work Section.
- We fixed all the typos spotted by Reviewer 2, included the minor improvements suggested
by Reviewer 4 and fixed the algorithm numbering as noted by Reviewer 3.
- We changed the caption of Figure 1, as suggested by Reviewer 3.

We are looking forward to your feedback, and we will continue working on the improvements
we promised (specifically improving the clarity of the figures, improving the Appendix, adding an
example to answer Reviewer 2's Question 2 ).

---

### Public Comment · ~Juntang_Zhuang1 · 2020-11-13
**Missing reference to a closely related work**

Hi, thanks for the nice work. We noticed that your work is closely related to the ICML paper [1], and would appreciate it if you could briefly discuss.

In the abstract, you said "If the trained model is supposed to be a flow generated from an ODE, it should be possible to choose another numerical solver with equal or smaller numerical error without loss of performance". To our knowledge, [1] is among the earliest to achieve this goal and have similar observation as in your work. Please see table 2 in the main paper of [1], and table 7 in the appendix of [1], where the ODE model is tested with different solvers without re-training and still achieve similar results.

Regarding the accuracy on Cifar10, do you have any idea why your reported accuracy is below 60% in Fig 5? [1] has achieved over 90% test accuracy. Is it because of the model or the training? Thanks much in advance.

[1] Zhuang, Juntang, et al. "Adaptive Checkpoint Adjoint Method for Gradient Estimation in Neural ODE." arXiv preprint arXiv:2006.02493 (2020).

---

> ### Author Response · Authors · 2020-11-17
> **Re: Juntang Zhuang**
>
> Thank you for your interest in our paper. Indeed the work you mention is related to our work, specifically the
> results in Table 7. We will add [1] to our discussion, thank you for the suggestion.
>
> The reason our model achieves much lower accuracy than [1] is due to the simple architecture of our model.
> For example the model in [1] consists of multiple ODE blocks, whereas our model consists of only a single ODE block.
>
>
> [1] Zhuang, Juntang, et al. "Adaptive Checkpoint Adjoint Method for Gradient Estimation in Neural ODE."
> arXiv preprint arXiv:2006.02493 (2020).

---

### Author Response · Authors · 2020-11-19
**Revision 2**

We have incorporated further improvements and uploaded a new draft of our paper.

These improvements include:
- We improved the layout of the figures by making them wider and adding
legends as suggested by Reviewer and Reviewer 4 and Reviewer 3.
Additionally, we reduced the amount of data shown in each figure to make them clearer.
- We added a descriptive text to the Supplementary Material and improved the
overall layout.
- We added a new section to the appendix in answer to Question 2 of Reviewer 2.
- We have added the suggested additional reference.
- Changed the title to: ResNet After All? Neural ODEs and Their Numerical Solution

We have started experiments, as suggested by Reviewer 4, and hope to
add additional experimental results soon.

---

### Author Response · Authors · 2020-11-22
**Revision 3**

As suggested by Reviewer 4 we have added additional experiments to our paper. These experiments are:

- Experiments on MNIST: Fixed step solver experiments, adaptive step size solver experiments, step adaptation algorithm
with Euler as train solver and Midpoint as test solver.
- Step adaptation algorithm with Euler as train solver and rk4 as test solver on CIFAR10 and Sphere2.
- Step adaptation algorithm with Midpoint as train solver and rk4 as test solver on CIFAR10 and Sphere2.

The experimental results can be found in the Supplementary Material Section B.

---

### Decision · Program_Chairs · 2021-01-07
**Final Decision**

**Decision:**

Accept (Poster)

**Comment:**

The paper considers whether Neural ODEs have a valid interpretation as an ODE, showing that such an interpretation is not correct unless the discretization is chosen properly.  This is important, given interest in Neural ODEs as models as well as they way they will be used, both for problems involving physical/temporal data as well as more generally.  The paper proposes an algorithm for adapting integration step-size during training to partially address this issue, and empirical results are shown.  There was a detailed discussion between reviewers and authors which led to improvements.  The authors should also discuss the relationship of their work with https://arxiv.org/abs/2008.02389, which makes a similar point, in the final version.